# Ariadne's Thread of LipSync: Unraveling Forgeries via Inconsistency between Lip Motions and Head Poses

**Tianyi She** [* 1]   **Jiawei Liu** [* 2]   **Weifeng Liu** [3]   **Hanqing Zhao** [4]   **Weiming Zhang** [1]   **Kejiang Chen** [1]

## Abstract

Recent advances in LipSync generation technology have led to the creation of highly realistic videos, posing severe societal risks. However, existing defense strategies struggle against LipSync forgeries, as advanced LipSync generation methods not only achieve better lip synchronization but also eliminate visual artifacts. An important reason is that they overlook an inherent biological coupling between lip movements and head poses in natural speech videos. In this paper, we propose LipDA, a novel framework for joint LipSync Detection and Attribution, which takes advantage of the inconsistency between head and lip. For detection, the framework learns to quantify this discrepancy by contrasting lip and pose features from authentic versus forged videos. For attribution, our method is designed to capture the unique temporal dynamics and audio-visual synchronization patterns that act as the fingerprint of models, enabling source tracing. We conduct extensive experiments on two challenging LipSync datasets as well as our own proposed large-scale and multi-generator dataset. LipDA achieves over 97% AUC in detection and 97.5% accuracy in model attribution, significantly outperforming existing methods.

## 1. Introduction

The rapid proliferation of powerful generative models (Goodfellow et al., 2014; Ho et al., 2020; Yang et al., 2023) has fueled the creation of synthetic multimodal content at an unprecedented scale. Within this landscape, LipSync generation has emerged as a transformative tool

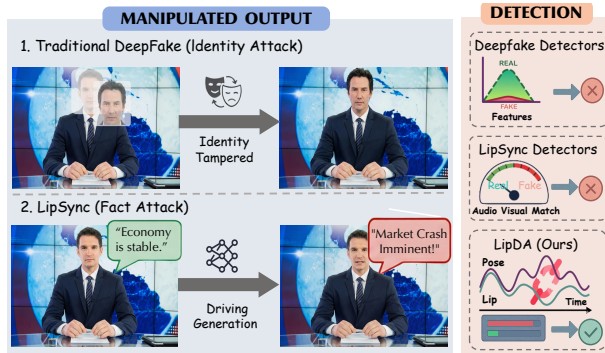

*Figure 1.* **Taxonomy of forgeries and detection comparison. (Left)** Distinct from identity attacks, LipSync constitutes a fact attack, fabricating videos where the target utters speech never spoken. **(Right)** Vulnerability of existing defenses versus our method. LipDA robustly identifies forgeries by capturing the inconsistencies between lip and head dynamics.

widely deployed in digital avatar creation and virtual anchoring (Gan et al., 2025; Meng et al., 2024). This technology operates by utilizing a reference audio or video driving signal to manipulate the lip region of a source identity, enforcing strict synchronization to synthesize realistic talking heads while preserving the speaker's original identity.

However, the widespread availability of open-source models has introduced severe security risks. Adversaries increasingly exploit LipSync for disinformation and financial fraud (Suwajanakorn et al., 2017), as evidenced by a recent incident in which criminals impersonated a corporate executive to defraud a company of $25 million (James et al., 2026). In contrast to traditional DeepFakes such as face swapping, attribute editing, and entire face synthesis (Xu et al., 2022; Yan et al., 2023), LipSync preserves the victim's authentic identity and visual context while fabricating speech that was never uttered, as illustrated in Fig. 1. This design renders it a more deceptive and stealthier form of forgery. Consequently, advanced defense frameworks are urgently needed to both detect these subtle forgeries and attribute them to their source models for forensic accountability.

Current defense strategies struggle with evolving LipSync generation. **First**, general DeepFake detectors (Tan et al., 2024b;a; Ojha et al., 2023), which primarily target spatial-

[1] University of Science and Technology of China, HeFei, China [2] Shanghai Jiaotong University [3] Peking University [4] Nanyang Technological University. Correspondence to: Kejiang Chen <chenkj@ustc.edu.cn>.

*Proceedings of the 43rd International Conference on Machine Learning*, Seoul, South Korea. PMLR 306, 2026. Copyright 2026 by the author(s).

frequency anomalies in identity-tampering scenarios, often falter in the LipSync domain. Since LipSync involves fine-grained, localized manipulations, these sparse forgery signals are easily diluted by overwhelming authentic facial features, leading to a loss of discriminative power. **Second**, specific LipSync detectors (Haliassos et al., 2022; Wang et al., 2023b; Smeu et al., 2025) relying on audio-visual inconsistencies or local artifacts are becoming ineffective. Recent diffusion-based generators synthesize high-fidelity textures while stabilizing lip dynamics. These advancements effectively mitigate the viseme-phoneme mismatches and visual artifacts that prior methods relied upon. **Finally**, existing literature largely overlooks the critical task of model attribution. Relying solely on binary classification (real vs. fake) is insufficient for real-world forensics, where identifying the specific source generator is paramount for accountability.

To address these challenges, we investigate the fundamental discrepancy between authentic speech and LipSync forgeries. In natural speech, lip articulation and head dynamics are inherently coupled, driven by a unified neuro-cognitive control system (Chen et al., 2020; Gick et al., 2020). For instance, prosodic emphasis deterministically coordinates with head keypoints. Although advanced models can successfully synthesize plausible head motion, they operate under a decoupled control paradigm. Specifically, the lip movement is subject to hard constraints from the audio input to ensure semantic accuracy, whereas the head pose is generated via probabilistic sampling from a learned distribution or transplanted from an uncorrelated driving source. This fundamental decoupling disrupts the intrinsic biological synchronization, resulting in subtle yet quantifiable lip-pose inconsistencies, which serve as the "Ariadne's Thread" for detecting forgeries. Furthermore, we observe that the distinct synthesis mechanisms of different generator families imprint unique temporal fingerprints on the video, providing a discriminative basis for source model attribution.

Inspired by above analysis, we propose a novel framework for joint **Lip**Sync forgeries **D**etection and **A**ttribution (LipDA). For detection, we introduce a lip-pose contrastive encoding mechanism. By projecting temporal lip features and head dynamics into a shared physiological manifold for detection classification, this module effectively discriminates natural biological coupling from the disrupted motion characteristic of forgeries. For attribution, we incorporate modality synchronization and temporal dynamic modules to capture the unique architectural fingerprints imprinted by distinct synthesis models via temporal motion patterns. Furthermore, to address the scarcity of high-quality forensic resources in this domain, we construct LipSync-A, providing fine-grained labels explicitly designed for attribution tasks. Extensive experiments validate LipDA's effectiveness, demonstrating superior cross-dataset generalization and robustness compared to state-of-the-art methods.

Our work makes the following key contributions:

- **Physiological Signal Unveiling.** We reveal that the intrinsic inconsistency between lip motion and global head pose is inherently disrupted by LipSync generation, and its distinct temporal dynamics serve as a reliable fingerprint for source model attribution.

- **Unified Detection and Attribution Framework.** We propose the first unified framework to jointly address LipSync detection and source attribution. By integrating contrastive learning with cross-modal temporal modeling, our two-stage design effectively captures discriminative generative patterns.

- **Superior Generalization and Robustness.** Extensive experiments demonstrate that LipDA significantly outperforms existing detectors. Crucially, it exhibits strong cross-domain generalization to unseen models and general deepfakes, while maintaining stability under challenging visual and audio perturbations.

## 2. Related Work

### 2.1. LipSync Generation

Early approaches relied on geometric transformations (Wiles et al., 2018) and template-based rules (Bouaziz et al., 2013), which were labor-intensive and often lacked visual fidelity. The advent of deep learning enabled more effective synthesis. Siarohin et al. (Siarohin et al., 2019) proposed a video-driven framework leveraging keypoint transformations to predict motion, though it yielded unstable results under large pose variations and occlusions. In contrast, audio-driven approaches focus on mapping audio features to visual synthesis. Seminal works like Wav2Lip (Prajwal et al., 2020) utilized GANs with expert discriminators to enforce precise LipSync, though often at the expense of pose diversity. To enhance controllability and realism, recent research has shifted towards sophisticated generative paradigms. Methods such as SadTalker (Zhang et al., 2023a) and VAE-based frameworks (Wang et al., 2024) employ intermediate representations to decouple motion from appearance. By modeling the probabilistic distribution of motion, diffusion-based models (Ma et al., 2023; Ji et al., 2025) generate highly expressive dynamics and high-fidelity textures. This effectively eliminates the jittering artifacts prevalent in earlier generations, thereby rendering traditional artifact-based detection cues increasingly ineffective.

### 2.2. LipSync Forgery Detection

Since LipSync manipulations require detectors to capture fine-grained localized dynamics, early general approaches targeting spatial artifacts (Wu et al., 2022) or frequency

anomalies (Tan et al., 2024a) often falter in this domain due to the subtle nature of lip modifications. Consequently, research has pivoted towards modeling temporal coherence and semantic consistency. Visual-only methods, such as LipForensics (Haliassos et al., 2021), exploit high-level semantic irregularities using pre-trained lip-reading networks, while RealForensics (Haliassos et al., 2022) and FTCN (Zheng et al., 2021) leverage self-supervised learning to detect temporal discontinuities in facial motion. More critically, considering the multimodal nature of LipSync, detecting audio-visual inconsistencies has become a dominant paradigm. Recent state-of-the-art methods leverage cross-modal dissonance as a primary forensic cue. For instance, AVAD (Feng et al., 2023) and SpeechForensics (Liang et al., 2024) learn joint audio-visual representations to spot synchronization anomalies. Similarly, LipFD (Liu et al., 2024) and AVH-align (Smeu et al., 2025) explicitly contrast visual lip dynamics with audio spectrograms to identify fine-grained mismatches. Nevertheless, these strategies face diminishing returns. As diffusion-based generators synthesize high-fidelity textures and strictly synchronized lip motions, the explicit artifacts and synchronization errors relied upon by prior methods are increasingly mitigated. Furthermore, existing literature remains largely confined to binary classification, overlooking the critical task of source model attribution necessary for forensic accountability.

## 3. Analysis of LipSync Forgery Signals

### 3.1. Inter-Class Signal: Real vs. Fake

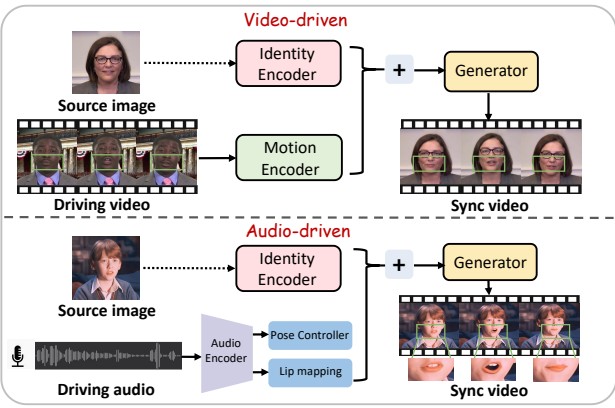

*Figure 2.* **Illustration of the two primary LipSync forgery paradigms**. Top (Video-driven): Motion and head pose are transplanted from a driving video, creating localized temporal artifacts. Bottom (Audio-driven): Lip motion is synthesized from audio, severing the global link to head motion, which remains static or is generated independently.

In natural speech, lip motion is inherently coordinated with surrounding facial muscle activity and is accompanied by slight head movements that correlate with speech

rhythm (Gick et al., 2020). We find that current LipSync forgeries fundamentally disrupt this physiological coupling, which is an inherent limitation of their synthesis mechanisms. We analyze the two primary paradigms, as illustrated in Fig. 2. Video-driven approaches directly learn and transplant motion from a driving video onto a source identity. This process inherently replaces the source's intrinsic motor patterns with the driver's head-pose and expression dynamics. The result is subtle temporal artifacts, as this foreign lip-head pattern is unnatural for the source identity.

Audio-driven methods commonly decompose LipSync generation into two independent sub-problems, namely an audio-to-lip mapping and a head-pose generation pathway. This decomposition manifests in different forms across architectures. Reconstruction-based approaches such as Wav2Lip mask the lower face and synthesize lip movements under lip-reading constraints, while retaining the head pose statically from the source frame. Diffusion-based approaches instead learn a dedicated head pose predictor that fuses audio features with trained image embeddings to produce motion parameters, which are then injected into the denoising process. By treating lip articulation and head pose as separable conditional distributions, they sever the intrinsic global coupling between the two parts. Furthermore, in contrast to the relatively well defined phoneme-to-viseme mapping, lip-head coordination is a complex, high-dimensional, and global task intrinsically tied to speaker identity. These inherent deficiencies in existing LipSync synthesis mechanisms provide crucial and exploitable cues for detection (see Appendix B.3 for more details).

We validate the aforementioned observations through the facial action unit (AU) analysis in Fig. 3. Video-driven forgeries correspond to significantly reduced intensity values, indicating that facial activity is suppressed by the influence of the driving signal. These transplanted foreign motion patterns result in unnatural lip trajectories. Conversely, while audio-driven forgeries successfully mimic authentic lip dynamics, they exhibit erratic peaks in AU17, revealing a failure to achieve lip-pose coordination.

### 3.2. Intra-Class Signal: Model Fingerprints

Existing data-driven attribution methods primarily rely on dataset-specific artifacts rather than intrinsic generative mechanisms, rendering them susceptible to training set bias. Therefore, discriminative intra-class signals inherent to the model architecture serve as a more reliable cue for the attribution task. Building on the concept of generative fingerprints (Song et al., 2024), we empirically hypothesize that LipSync model families possess such fingerprints in their temporal dynamics and motion patterns. This is because different generative architectures employ distinct approaches to modeling temporal coherence, such as adversarial processes

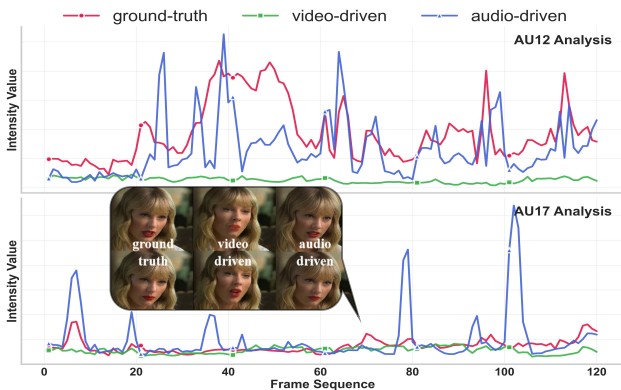

*Figure 3.* **AU intensity analysis on original video sequences and two categories of forgery patterns**. Higher intensity values indicate heightened activity of specific facial muscles. Two representative action units, AU12 (mouth corner pull) and AU17 (chin raise), are analyzed.

or latent reconstruction, which impose unique statistical constraints on the resulting motion patterns.

To validate this hypothesis, we sample forged videos generated by different model families using disjoint source identities and diverse driving signals. For feature extraction, we first track facial landmarks across the frame sequence and compute the 6-Degrees-of-Freedom (6-DoF) head pose vector sequence. This raw motion sequence is then encoded by a spatial-temporal transformer to extract feature embeddings. Finally, we apply t-SNE to visualize the distribution of these embeddings. As shown in Fig. 4, the results clearly reveal well-separated clusters, strongly suggesting that the temporal dynamics of inter-frame head motion indeed provide a separable feature space for model attribution.

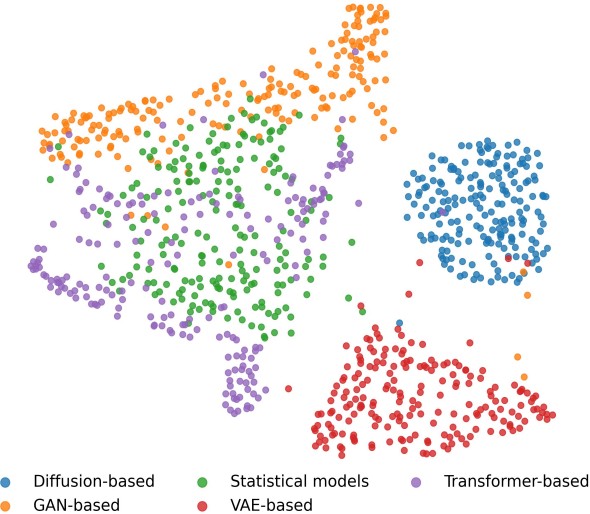

● Diffusion-based    ● Statistical models    ● Transformer-based
● GAN-based          ● VAE-based

*Figure 4.* **t-SNE visualization of pose features from LipSync forgeries generated by different model families**.

### 3.3. Dataset construction

Existing datasets often cover only a limited number of generators or rely on outdated synthesis pipelines. To bridge this gap, we construct LipSync-A, the first large-scale dataset explicitly designed for LipSync forensics. As shown in Table 1, it encompasses 15 SOTA generators spanning 7 architectures, providing 16,000 synthetic videos for LipSync research. Crucially, every video in LipSync-A is meticulously labeled with its source generator, enabling us to evaluate the attribution performance. The overall data-generation pipeline and comprehensive details on the selected generators and dataset splits are provided in the Appendix A.

*Table 1.* **Comparison with existing LipSync datasets. #Gens.** and **#Fakes** denote the number of generator architectures and forged video samples, respectively.

| Dataset | Detection | Attribution | # Gens. | # Fakes |
|---|---|---|---|---|
| TalkHeadBench(Xiong et al., 2025) | ✓ | ✗ | 2 | 2,984 |
| AVLips (Liu et al., 2024) | ✓ | ✗ | 2 | 4,206 |
| LAV-DF (Cai et al., 2022) | ✓ | ✗ | 1 | 99,873 |
| PolyGlotFake (Hou et al., 2024) | ✓ | ✗ | 2 | 14,472 |
| AV-Deepfake1M (Cai et al., 2024) | ✓ | ✗ | 1 | 860,039 |
| LipSync-A(Ours) | ✓ | ✓ | 7 | 16,000 |

## 4. Methodology

In this section, we introduce a unified two-stage framework designed for joint LipSync detection and source attribution, as illustrated in Fig. 5. Stage I employs a contrastive mechanism to capture lip-pose inconsistencies for robust binary detection, while Stage II integrates audio-visual synchronization (MSM) and temporal motion patterns (TDM) to disentangle fine-grained model fingerprints for attribution.

### 4.1. Problem Formulation

Let the dataset $\mathcal{D}$ consist of real videos $x_{\text{real}}$ and LipSync-generated forgeries $x_{\text{fake}}$. Given an input video $x^{\text{video}}$, we extract the frame sequence $\{x_1^i, \ldots, x_m^i\}$ and audio signal $x^a$. The detection task is a binary classification problem:

$$y^{\text{video}} = f_{\text{det}}(x^{\text{video}}) \in \{real : 0, fake : 1\}. \quad (1)$$

The objective of Stage I is to train a robust detection classifier $f_{\text{det}}(\cdot)$. For attribution, we define a candidate generator set $\mathcal{M} = \{\text{model}_1, \text{model}_2, \ldots, \text{model}_n\}$, and aim to identify the source model $\hat{m}$ for a forged sample:

$$\hat{m} = f_{\text{att}}(x^{\text{video}}) \in \mathcal{M}. \quad (2)$$

### 4.2. Stage I: Pose-Lip Contrastive Optimization

**Pose and Lip Encoders.** Stage I aims to capture the intrinsic consistency between head poses and lip motions that is often disrupted in forgeries. We first adopt a sliding window

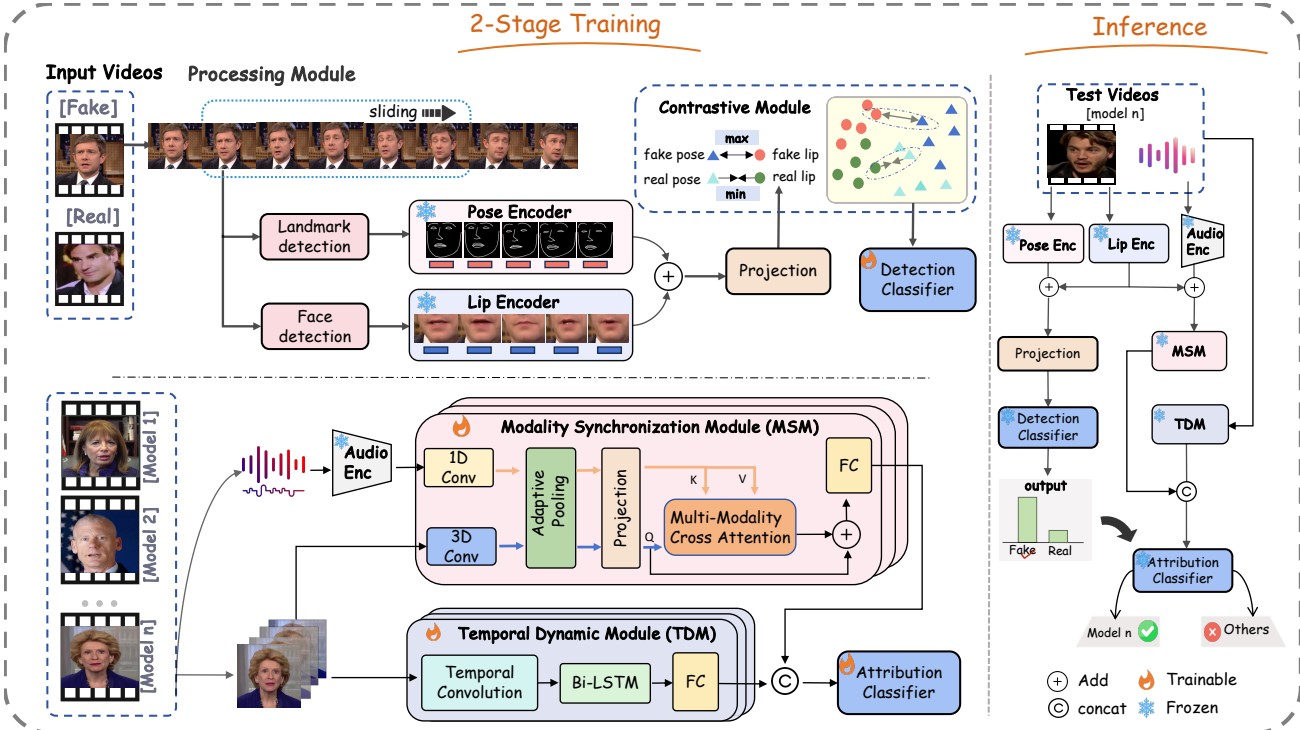

*Figure 5.* **Overview of our unified LipDA framework**. 2-Stage Training (Left): In Stage I Detection, a dual-branch encoder is trained with a contrastive module to learn the lip-head pose inconsistency. In Stage II Attribution, Stage I encoders are frozen, and the Modality Synchronization (MSM) and Temporal Dynamic (TDM) modules are trained to capture unique generative fingerprints. Inference (Right): The unified pipeline integrates all components for joint forgery detection and source attribution.

of $T$ frames to form video clips. Since LipSync manipulations primarily affect the lower facial region, we first extract the lip region $L = \{l_i \in \mathbb{R}^{3 \times H_{\text{lip}} \times W_{\text{lip}}}\}_{i=1}^{T}$ from each frame. These patches are processed by a ResNet-based encoder $f_{\text{res}}(\cdot)$. The resulting per-frame embeddings are concatenated, and projected into a latent space $\mathbb{R}^o$ via a dedicated projection head $g_l(\cdot)$, yielding the final lip embedding.

$$O_l = g_l(\text{Concat}(f_{\text{res}}(l_i))). \tag{3}$$

To reduce sensitivity to pixel-level variations, we represent head motion via facial landmark trajectories $P = \{p_i \in \mathbb{R}^{468 \times 3}\}_{i=1}^{T}$. This sequence is processed by a separate pose projection head $g_p(\cdot)$ to embed it into the same latent space:

$$O_p = g_p(P) \in \mathbb{R}^o. \tag{4}$$

**Contrastive and Detection Objectives.** We aim to learn a shared latent space $\mathbb{R}^o$ where the lip embedding $O_l$ and pose embedding $O_p$ are discriminative. For authentic videos ($y = 0$), these embeddings should be aligned, whereas forgeries ($y = 1$) should diverge. To enforce this property, we employ a margin-based contrastive loss $\mathcal{L}_{\text{align}}$. This objective explicitly minimizes the $L_2$ discrepancy for real samples while penalizing forged pairs that fall within a

margin $\gamma$.

$$\begin{aligned} \mathcal{L}_{\text{align}} = & \mathbb{E}_{y=0}\left[\|O_p - O_l\|_2^2\right] \\ & + \mathbb{E}_{y=1}\left[\max\left(0, \gamma - \|O_p - O_l\|_2^2\right)\right]. \end{aligned} \tag{5}$$

To achieve alignment, an MLP is trained on the concatenated features $\text{Concat}(O_p, O_l)$. It is optimized with a standard binary cross-entropy loss:

$$\mathcal{L}_{\text{det}} = -\mathbb{E}\left[y \log(\hat{y}) + (1 - y) \log(1 - \hat{y})\right]. \tag{6}$$

The final Stage I objective is a joint optimization of both losses, and $\lambda_{\text{align}}$ is a balancing hyperparameter.

$$\mathcal{L}_{\text{stage1}} = \lambda_{\text{align}}\mathcal{L}_{\text{align}} + \mathcal{L}_{\text{det}}. \tag{7}$$

### 4.3. Stage II: Cross-Modal Temporal Modeling

While Stage I distinguishes real from fake, Stage II is designed for attribution, identifying which generative model created a forgery. This stage uses two dedicated modules TDM and MSM, which are trained independently from Stage I but used concurrently during inference. To capture unique temporal fingerprints, we extract three feature streams from the input video: The facial keypoint sequence $\mathbf{K} \in \mathbb{R}^{T \times k \times 3}$, the lip ROI sequence $\mathbf{L} \in \mathbb{R}^{T \times c \times h \times w}$, and

the aligned MFCC features $\mathbf{A} \in \mathbb{R}^{T \times d_a}$. Here, $T$ is the sequence length; $k$ is the number of keypoints; $c, h, w$ are the channels, height, and width of the lip ROIs; and $d_a$ is the MFCC feature dimension.

**Temporal Dynamic Module (TDM).** To capture generator-specific motion artifacts, the TDM module encodes the key-point sequence $\mathbf{K}$. It employs a Temporal-CNN to learn local motion patterns, followed by a Bi-LSTM to model long-range dependencies, yielding the temporal feature $z_{\text{temp}}$:

$$z_{\text{temp}} = f_{\text{BiLSTM}}(\text{Conv}(\mathbf{K})). \qquad (8)$$

**Modality Synchronization Module (MSM).** To estimate audio-visual alignment, we encode the lip ROIs $\mathbf{L}$ and audio features $\mathbf{A}$ using dedicated 3D-CNN and 1D-CNN encoders. Both resulting feature maps are then processed by an adaptive pooling layer and subsequent projection head to generate the final visual ($L'$) and audio ($A'$) representations. We then apply multi-head cross-attention.

$$L' = f_{\text{3D}}(\mathbf{L}) \in \mathbb{R}^{T \times d_v}, A' = f_{\text{1D}}(\mathbf{A}) \in \mathbb{R}^{T \times d_a} \qquad (9)$$
$$\tilde{L} = \text{Attn}(Q = L', K = A', V = A').$$

Finally, the attended features $\tilde{L}$ are fused with the original visual features $L'$, aggregated by temporal averaging, and projected by $g_{\text{av}}$ to produce the synchronization feature:

$$z_{\text{av}} = g_{\text{av}}\left(\frac{1}{T}\sum_{t=1}^{T}[L'_t + \tilde{L}_t]\right). \qquad (10)$$

**Attribution Classifier.** We concatenate the temporal dynamic feature $z_{\text{temp}}$ and the synchronization feature $z_{\text{av}}$ to form a comprehensive fingerprint $z_{\text{final}}$. This classifier is trained exclusively on forged samples using their ground-truth model label $\hat{m} \in \mathcal{M}$. The objective is a standard cross-entropy loss $\mathcal{L}_{\text{att}}$:

$$\mathcal{L}_{\text{att}} = -\mathbb{E}\left[\log\left(f_{\text{att}}(z_{\text{final}})\right)_{\hat{m}}\right]. \qquad (11)$$

### 4.4. Inference

As shown in Fig. 5, all components form a unified pipeline during the inference stage. The Detection Classifier ($f_{\text{det}}$) predicts the forgery label $\hat{y} \in \{\text{Real, Fake}\}$. Concurrently, features from the MSM and TDM modules are concatenated and passed to the Attribution Classifier ($f_{\text{att}}$) to predict the source model label $\hat{m} \in \mathcal{M}$. The pipeline outputs a joint prediction $(\hat{y}, \hat{m})$ for simultaneous detection and attribution.

## 5. Experiment

### 5.1. Experimental Setup

**Dataset and Evaluation Protocols.** We utilize our proposed LipSync-A as the primary dataset for training and in-domain detection evaluation, leveraging its diverse generator architectures to enable fine-grained attribution evaluation. To rigorously assess generalization, we adopt a zero-shot evaluation protocol on AVLips and TalkHeadBench to test performance across unseen data distributions, while further verifying adaptability against novel SOTA synthesis algorithms excluded from training. Additionally, we examine cross-manipulation transferability using the Deep-Fake dataset Celeb-DF (Li et al., 2020), whereas robustness against diverse visual and audio perturbations is assessed on AVLips, adhering to standard forensic protocols.

**Metrics.** Following standard practices (Xiong et al., 2025; Liang et al., 2024), we report video-level Accuracy (ACC) and Area Under the ROC Curve (AUC) for comparison. We also include F1-score, Average Precision (AP), False Positive Rate (FPR), and False Negative Rate (FNR) for a comprehensive evaluation.

**Baselines and Implementation Details.** We compare against 15 SOTA deepfake detectors, including 8 uni-modal approaches that cover lip-reading, temporal coherence, and frequency-domain cues, and 7 multi-modal approaches that exploit audio-visual asynchrony or alignment. For LipDA, we extract facial landmarks with MediaPipe, set the sliding-window length to $T = 5$ frames, and crop lip ROIs at $96 \times 96$. Audio is encoded as MFCC features aligned to the visual stream. Both stages are optimized with Adam at an initial learning rate of $1 \times 10^{-5}$, batch size 32, for 20 epochs. Further hyperparameters and training composition are detailed in Appendix C.

### 5.2. Comparison to Existing Methods

As shown in Table 2, our method LipDA significantly outperforms existing baselines on three LipSync datasets in both video-only and audio-visual settings. In the audio-visual setting, it surpasses the previous state-of-the-art SpeechForensics by +7.46% in ACC and +7.97% in AUC. We observe that many unimodal detectors perform near random guessing, indicating their inability to handle subtle LipSync forgeries. LipForensics and RealForensics utilizing lip-shape cues achieve better results, but they are still substantially outperformed by multimodal methods. While incorporating audio generally improves our model's performance, we note a slight AUC decrease on TalkHeadBench. We hypothesize that this is due to the high variability introduced by the real-scene audio in that specific dataset. Table 3 presents the performance of our method compared with advanced multimodal detectors on the attribution task across five generator families. Our model demonstrates highly balanced recognition, achieving an average F1-score improvement of over 12 percentage points compared to the second-best method, TALL. While SpeechForensics and AVAD exhibit severe instability with extremely low F1-scores on the GAN

*Table 2.* **Binary forgery detection performance comparison**. We report Accuracy (ACC, %) and AUC (%) on three public benchmarks against SOTA methods. 'V' denotes our visual-only model and 'A-V' denotes our audio-visual variant, which additionally incorporates the MSM score for detection. Bold indicates the best results, while the second-ranking one is underscored.

| Method | Modality | In-domain LipSync-A | | Cross-domain AVLips | | TalkHeadBench | | Average | |
|---|---|---|---|---|---|---|---|---|---|
| | | ACC ↑ | AUC ↑ | ACC ↑ | AUC ↑ | ACC ↑ | AUC ↑ | ACC ↑ | AUC ↑ |
| FTCN (Zheng et al., 2021) | V | 78.98 | 87.78 | 65.67 | 71.55 | 68.73 | 75.46 | 71.13 | 78.26 |
| CADDM (Dong et al., 2023) | V | 47.15 | 63.14 | 45.90 | 55.92 | 62.53 | 63.69 | 51.86 | 60.92 |
| LipForensics (Haliassos et al., 2021) | V | 80.92 | 89.85 | 74.15 | 81.97 | 84.52 | 90.63 | 79.86 | 87.48 |
| RealForensics (Haliassos et al., 2022) | V | 72.44 | 67.03 | 80.47 | 89.25 | 71.03 | 79.01 | 74.65 | 78.43 |
| UnivFD (Ojha et al., 2023) | V | 52.30 | 69.27 | 50.00 | 47.34 | 56.90 | 76.25 | 53.07 | 64.29 |
| FreqNet (Tan et al., 2024a) | V | 48.80 | 9.40 | 45.30 | 52.40 | 42.10 | 73.60 | 45.40 | 45.13 |
| NPR (Tan et al., 2024b) | V | 45.6 | 14.40 | 44.40 | 44.00 | 54.10 | 74.10 | 48.03 | 44.17 |
| TALL (Xu et al., 2024b) | V | 75.24 | 96.92 | 61.36 | 80.69 | 58.88 | 71.49 | 65.16 | 83.03 |
| AltFreezing (Wang et al., 2023b) | A-V | 62.75 | 87.09 | 50.16 | 70.01 | 50.29 | 64.11 | 54.40 | 73.74 |
| AVAD (Feng et al., 2023) | A-V | 48.72 | 24.81 | 71.00 | 73.18 | 58.20 | 58.66 | 59.31 | 52.22 |
| LipFD (Liu et al., 2024) | A-V | 50.45 | 44.09 | 95.27 | 96.08 | 43.82 | 44.75 | 63.18 | 61.64 |
| SpeechForensics (Liang et al., 2024) | A-V | 91.38 | 94.82 | **98.50** | 99.15 | 76.52 | 78.87 | 88.80 | 90.95 |
| FGMDC (Yin et al., 2024) | A-V | 63.09 | 60.00 | 84.00 | 92.34 | 89.78 | 90.80 | 78.96 | 81.05 |
| DFD-FCG (Han et al., 2025) | A-V | 83.26 | 88.34 | 66.33 | 68.27 | 64.75 | 66.55 | 71.45 | 74.39 |
| AVH-align (Smeu et al., 2025) | A-V | 94.17 | 98.16 | 75.74 | 88.38 | 45.80 | 34.95 | 71.90 | 73.83 |
| LipDA(Ours) | V | 93.47 | 97.83 | 97.02 | 99.59 | 93.05 | **98.65** | 94.51 | 98.69 |
| LipDA(Ours) | A-V | **95.96** | **99.42** | 98.34 | **99.82** | **94.48** | 97.50 | **96.26** | **98.91** |

*Table 3.* **Attribution performance comparison for classifying forgeries into five generator families**. Metrics are Accuracy (ACC, %) and F1-Score (F1, %). All listed methods are audio-visual. Bold and underscored denote the best and second-best results, respectively.

| Method | Transformer-based | | GAN-based | | Diffusion-based | | VAE-based | | Statistical models | | Average | |
|---|---|---|---|---|---|---|---|---|---|---|---|---|
| | ACC ↑ | F1 ↑ | ACC ↑ | F1 ↑ | ACC ↑ | F1 ↑ | ACC ↑ | F1 ↑ | ACC ↑ | F1 ↑ | ACC ↑ | F1 ↑ |
| AVAD (Feng et al., 2023) | 77.2 | 9.9 | 83.0 | 56.7 | 70.0 | 27.4 | 68.8 | 40.2 | 71.9 | 33.0 | 74.2 | 33.4 |
| TALL (Xu et al., 2024b) | 89.0 | 59.3 | 92.0 | 83.0 | 97.8 | 96.7 | 96.0 | 96.6 | 86.7 | 72.0 | 92.3 | 81.5 |
| AVH-align (Smeu et al., 2025) | 71.8 | 41.0 | **96.5** | 69.2 | **99.9** | 98.5 | 96.9 | 90.9 | **99.9** | 88.9 | 93.0 | 77.7 |
| SpeechForensics (Liang et al., 2024) | 70.4 | 43.9 | 71.6 | 18.4 | 70.4 | 46.0 | 76.4 | 11.9 | 73.2 | 13.0 | 72.4 | 26.6 |
| LipDA(Ours) A-V | **93.8** | **87.6** | 94.9 | **86.7** | **99.9** | **99.9** | **99.9** | **99.9** | 98.9 | **95.2** | **97.5** | **93.9** |

and Transformer categories, our model maintains robust performance even on the most challenging Transformer-based forgeries.

## 5.3. Generalizability to Unseen Forgery

We evaluated the generalization capability of our method on forgeries generated by recent LipSync techniques, as well as on the traditional cross-manipulation Celeb-DF dataset. As Celeb-DF lacks audio, all methods were evaluated in a video-only setting for this test. As shown in Table 4, existing multimodal detectors suffer a severe performance collapse on the unseen LipSync samples. In contrast, our method achieves the highest ACC and F1 scores, significantly outperforming the strong SpeechForensics baseline. Furthermore, LipDA remains highly effective against traditional forgeries in Celeb-DF, demonstrating that physiological inconsistencies serve as reliable artifacts beyond the LipSync domain. As shown in Table 5, despite being trained exclusively on LipSync forgeries, LipDA successfully generalizes to identity-swap manipulations, achieving an ACC

of 92.76% and an FNR of only 1.54%.

*Table 4.* **Cross-manipulation generalization on LipSync (A-V).** Unseen LipSync generalization on Sonic (Ji et al., 2025), KDTalker (Yang et al., 2025b), and OmniSync (Peng et al., 2025).

| Method | Sonic | | KDTalker | | OmniSync | |
|---|---|---|---|---|---|---|
| | ACC ↑ | F1 ↑ | ACC ↑ | F1 ↑ | ACC ↑ | F1 ↑ |
| LipFD (Liu et al., 2024) | 48.6 | 7.1 | 45.6 | 6.9 | 51.2 | 5.8 |
| AVAD (Feng et al., 2023) | 49.6 | 66.3 | 49.7 | 66.4 | 49.5 | 66.2 |
| SpeechForensics (Liang et al., 2024) | 80.8 | 80.4 | 83.9 | 81.0 | 90.0 | 90.2 |
| DFD-FCG (Han et al., 2025) | 66.4 | 61.9 | 72.2 | 69.4 | 52.1 | 14.7 |
| AVH-align (Smeu et al., 2025) | 70.0 | 56.9 | 71.9 | 64.9 | 72.9 | 9.5 |
| AltFreezing (Wang et al., 2023b) | 49.6 | 66.6 | 59.3 | 69.2 | 33.0 | 50.0 |
| LipDA(Ours) | **87.8** | **84.3** | **84.6** | **84.3** | **95.5** | **95.4** |

## 5.4. Robustness Evaluation

To evaluate the feasibility of our model for real-world deployment, we assess its robustness against diverse visual and audio perturbations. As shown in Fig. 6, the performance of baseline methods deteriorates significantly under distortions such as Gaussian blur, compression, and contrast

*Table 5.* **Cross-manipulation generalization on DeepFake (V)**. Generalization performance on CelebDF (Li et al., 2020).

| Method | CelebDF | | | | |
| --- | --- | --- | --- | --- | --- |
| | ACC ↑ | AUC ↑ | AP ↑ | FPR ↓ | FNR ↓ |
| CADDM (Dong et al., 2023) | 53.83 | 81.44 | 95.70 | 9.55 | 51.95 |
| UnivFD (Ojha et al., 2023) | 50.23 | 66.39 | 67.96 | 1.00 | 99.00 |
| FreqNet (Tan et al., 2024a) | 49.80 | 53.50 | 51.11 | 4.00 | 99.90 |
| NPR (Tan et al., 2024b) | 50.10 | 46.80 | 47.21 | 0.00 | 100.0 |
| RealForensics (Haliassos et al., 2022) | 54.05 | 67.08 | 68.81 | 81.23 | 49.31 |
| LipForensics (Haliassos et al., 2021) | 71.91 | 80.83 | 81.01 | 29.21 | 26.87 |
| LipDA(Ours) | **92.76** | **91.86** | **98.38** | 4.33 | **1.54** |

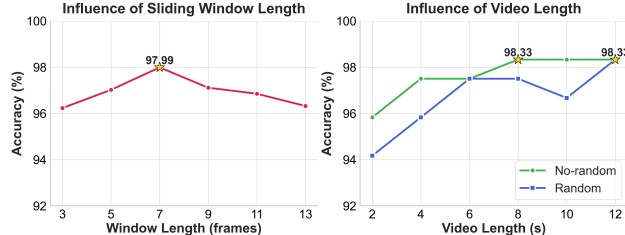

*Figure 7.* **Analysis of temporal parameters**. Left: Detection accuracy as a function of sliding window length $T$. Right: Detection accuracy by video length, comparing Random and Non-random sampling strategies.

adjustments. In contrast, LipDA exhibits remarkable stability against these texture-level corruptions. This resilience serves as strong evidence that our Stage I framework successfully captures the intrinsic physiological coupling between pose and lip dynamics, rather than overfitting to low-level artifacts. Furthermore, our method also demonstrates strong robustness to audio distortions. While sensitive to direct signal corruption, common operations like resampling and time shifts result in negligible performance drops. Full results are detailed in Appendix D (Table 10).

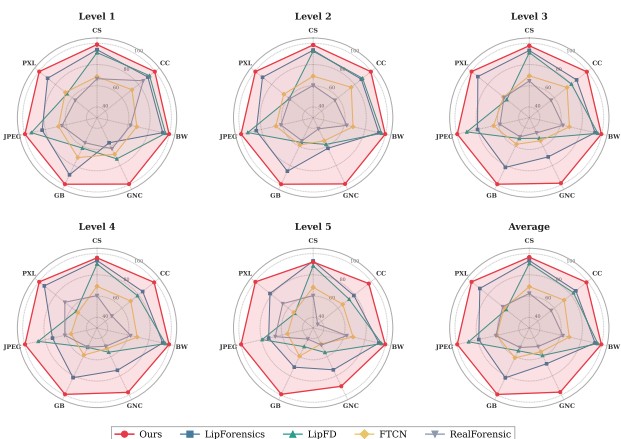

*Figure 6.* **Robustness against various unseen corruptions**. See Appendix D for detailed analysis and intensity settings.

### 5.5. Ablation Study and Analysis

**Influence of Sliding Window and Video Length**. As shown in Fig. 7, the sliding window length $T$ presents a clear trade-off. The window must be sufficiently long to capture meaningful temporal information, yet an excessively long window introduces redundant motion, which dilutes the critical inconsistency cues and degrades performance. Furthermore, detection accuracy improves with total video length, saturating at approximately 8 seconds under the Non-random sampling protocol.

**Effect of different components and backbone choices**. We conduct ablations to validate our component and backbone choices in Table 6. Removing the alignment loss leads

to a 17.3% drop, indicating that naive feature concatenation is insufficient and that contrastive alignment is necessary to capture lip–pose inconsistency between real and forged videos. Eliminating the lip component causes a performance collapse in detection, confirming that the mouth region provides the dominant forgery cues. For backbones, LSTM proves superior for pose features, while replacing ResNet-18 with CLIP-ViT reduces performance by 12.0%, likely due to a domain mismatch between large-scale pretraining and our small, constrained lip-crop domain. See Appendix F.4 for further attribution ablations.

*Table 6.* **Ablation study of our framework's components and backbone choices.** We report results on AVLips, and $\Delta$ denotes the ACC (%) drop from our full model.

| I. Component Removal | | | II. Backbone Replacement | | |
| --- | --- | --- | --- | --- | --- |
| Setting | ACC (%) | $\Delta$ (%) | Setting | ACC (%) | $\Delta$ (%) |
| Full model | **97.90** | — | Pose(Pooling) | **97.20** | ↓ 0.7 |
| w/o Align | 80.56 | ↓ 17.3 | Pose(Flatten) | 95.27 | ↓ 2.6 |
| w/o Pose | 74.78 | ↓ 23.1 | Lip(MobileNet) | 96.23 | ↓ 1.7 |
| w/o Lip | 53.50 | ↓ 44.4 | Lip(CLIP-ViT) | 85.73 | ↓ 12.0 |

**Visualization evidence for the proposed components**. We validate the Stage I framework by visualizing its learned feature space in Fig. 8. Lip and pose embeddings from unseen real samples are tightly intermingled, reflecting their inherent physiological coupling. Conversely, embeddings from fake samples form distinct, separable clusters, demonstrating that our model successfully learns to separate the inconsistent pose-lip features found in forgeries.

## 6. Discussion

Our work shifts LipSync detection from relying on local artifacts or explicit audio-visual mismatches to global physiological coordination. An adversary can easily optimize a lip-audio synchronization loss, but modeling the high-dimensional coordination of head motion with speech is substantially harder. Moreover, the robustness of our method to texture degradations confirms that it exploits these fundamental physiological constraints rather than brittle visual

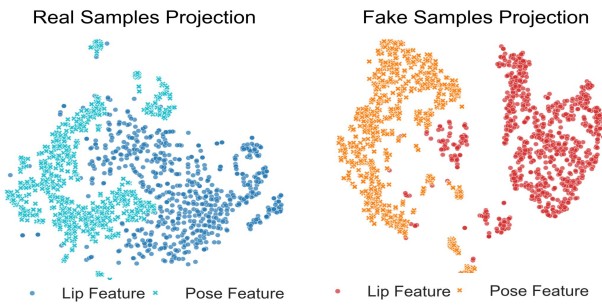

*Figure 8.* **t-SNE visualization of Stage I lip and pose embeddings from unseen real (Left) and fake (Right) test samples**.

cues. Despite the promising performance, LipDA has limitations that suggest future directions: (i) Scope. The applicability of this framework to broader AI-generated image forgery domains remains unexplored. (ii) Audio sensitivity. The Stage-II attribution module degrades under certain audio perturbations (Appendix D), indicating that practical applications should incorporate audio-quality awareness. (iii) Operational boundaries. Detection relies on visible lip dynamics and head motion, and degrades under stationary speakers, persistent downward gaze, or prolonged facial occlusion (Appendix F.4), motivating future work on complementary cues from upper-face or full-body dynamics.

## 7. Conclusion

We present a novel and robust forgery signal, the global physiological inconsistency between lip movements and head poses. We introduce LipDA, a two-stage framework that models this coordination to unify forgery detection and source attribution, which achieves superior performance, outperforming existing methods on both tasks. It also demonstrates strong generalization to unseen techniques and maintains high robustness against significant compression and blurring. Our research offers a more resilient defense mechanism, opening up a new path for the ongoing cat-and-mouse game of forgery generation and detection.

## Acknowledgements

This work was supported by National Natural Science Foundation of China (Grants 62372423, U2336206 and U2436601) and was also supported by the Fundamental Research Funds for the Central Universities WK2100250070.

## Impact Statement

This paper presents work whose goal is to advance the field of Machine Learning. There are many potential societal consequences of our work, none of which we feel must be specifically highlighted here.

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

# A. Additional Dataset Details

To address the critical scarcity of high-quality, diverse, and attribution-focused datasets for lip-synchronization (LipSync), we introduce LipSync-A, a comprehensive evaluation dataset designed to systematically cover dominant driving paradigms and identity input sources. It accommodates both audio- and video-driven generation paradigms, accepting inputs ranging from static images to dynamic source videos, thereby establishing a rigorous platform for attribution-oriented LipSync evaluation.

## A.1. Source Data Curation and Pre-processing

For identity inputs, we sourced data from LaPa (Liu et al., 2020), a large-scale in-the-wild facial image dataset covering rich expressions, poses, and occlusions; and HDTF (Zhang et al., 2021), a high-resolution talking head video dataset, mostly featuring English speakers in frontal views. For driving signals, we utilized VFHQ (Xie et al., 2022), a high-definition interview video dataset from 20 countries, covering natural blinking, speech, and large-scale head movements; and VCTK (Yamagishi et al., 2019), which provides 44.1 kHz clean read-speech sentences from 110 English speakers. We implemented a rigorous quality control pipeline, including automated filtering of low-quality samples, such as excessive blur, extreme poses, and uniform denoising of all audio tracks, as shown in Fig. 9.

## A.2. Forgery Generation Pipeline

A core strength of LipSync-A lies in its diverse and representative collection of generators. We incorporate both open-source state-of-the-art methods from academia and commercial APIs, encompassing a broad spectrum of technical paradigms. Following the taxonomy proposed by Meng et al. (2024), these include statistical models, hybrid transformations, and approaches based on keypoints, GANs, VAEs, diffusion models, and Transformers. In total, 15 distinct generation techniques were selected (refer to Table 7), yielding approximately 16,000 video samples. To bolster real-world robustness, we specifically generated approximately 270 samples using commercial APIs. For experimental evaluation, we employ a subset of this dataset partitioned with strict constraints to ensure disjoint identities and driving signals. This split results in 3,926 training, 841 validation, and 842 testing samples, guaranteeing a balanced representation across diverse technical methodologies and input conditions.

*Table 7.* **LipSync-A Generator Details.** Category refers to the core technical architecture. Paradigm indicates the driving signal. Original Training data lists the datasets used to train the original generator models.

| Generator | Category | Paradigm | Training Data |
|---|---|---|---|
| X2Face (Wiles et al., 2018) | Geometric | Video-driven | VoxCeleb |
| TPSM (Zhao & Zhang, 2022) | Landmark | Video-driven | VoxCeleb |
| LIA (Wang et al., 2022) | Latent Space | Video-driven | VoxCeleb |
| FaceVid (Wang et al., 2021) | Landmark | Video-driven | VoxCeleb |
| DiNet (Zhang et al., 2023b) | CNN | Audio-driven | HDTF, MEAD |
| MakeItTalk (Zhou et al., 2020) | GAN | Audio-driven | VoxCeleb, ObamaSet |
| Wav2Lip (Prajwal et al., 2020) | GAN | Audio-driven | LRW, LRS2 |
| TalkLip (Wang et al., 2023a) | GAN | Audio-driven | LRS |
| SadTalker (Xu et al., 2024b) | VAE | Audio-driven | VoxCeleb |
| IP_LAP (Zhong et al., 2023) | Transformer | Audio-driven | LRS2 |
| DreamTalk (Ma et al., 2023) | Diffusion | Audio-driven | MEAD, HDTF |
| Sonic (Ji et al., 2025) | Diffusion | Audio-driven | VFHQ, CelebV-Text |
| KDtalker (Yang et al., 2025a) | Diffusion | Audio-driven | HDTF |
| OmniSync (Peng et al., 2025) | Diffusion | Audio-driven | Web-Collected |
| InfiniteTalk (Yang et al., 2025c) | Diffusion | Audio-driven | Internal |

# B. Additional Discussion on Motivation

To address the theoretical underpinnings of our framework, we provide a deeper analysis of the fundamental constraints in LipSync generation and the discriminative basis for our attribution taxonomy.

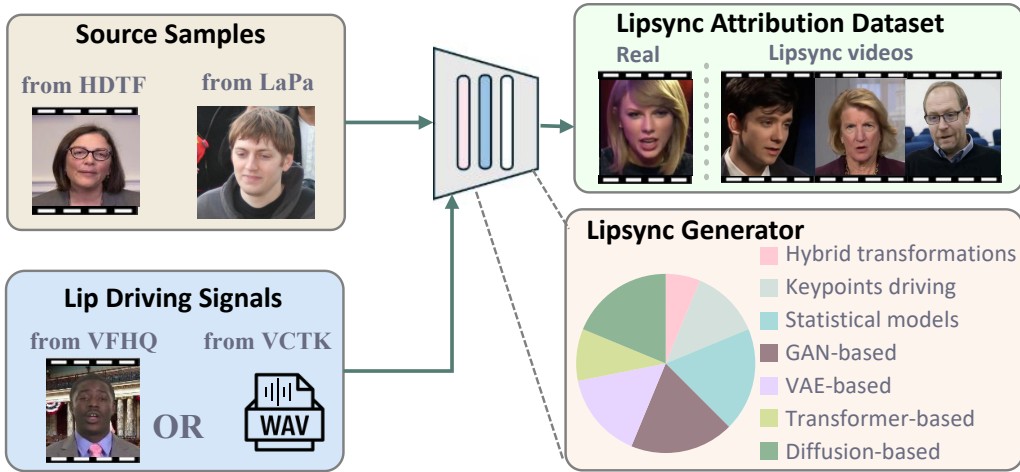

*Figure 9.* **The generation pipeline for our LipSync-A dataset**.

## B.1. Detection Motivation

One might question why the inconsistency between lip motion and head pose persists as a detectable signal. We argue that this is due to the inherent difficulty of simultaneously manipulating both modalities while maintaining physiological consistency.

**The Theoretical Gap**. In natural human speech, lip articulation ($L$) and head pose ($P$) are biologically coupled, driven by a unified neuro-cognitive system conditioned on the audio ($A$) and speaker identity ($I$). Theoretically, a perfect generator seeks to approximate the true joint distribution:

$$P(L, P | A, I)$$

However, modeling this high-dimensional joint distribution presents a significant optimization challenge due to the discrepancy in mapping properties. The mapping $\{A, I\} \rightarrow L$ is largely deterministic and strictly constrained by phoneme-to-viseme consistency for semantic intelligibility. In contrast, the mapping $\{A, I\} \rightarrow P$ is high-dimensional and one-to-many, as head motion is not governed by explicit phonetic content.

To reduce optimization difficulty and avoid mode collapse, current SOTA methods predominantly decompose the problem into two independent sub-problems:

$$P(L, P \mid A, I) \approx P(L \mid A, I) \cdot P(P \mid A, I)$$

By treating these as independent variables, generators inherently lose the correlation information between the specific phoneme articulation and the subtle head micro-movements. While lip movements are synchronized with audio, head motion is generated via independent probabilistic sampling or reference-driven conditioning, the correlation constraints between the two modalities are inherently absent. Our LipDA framework is explicitly designed to exploit this missing conditional dependence in the generative model.

## B.2. Attribution Motivation

To the best of our knowledge, we propose the first attribution task that taxonomizes audio-driven LipSync forgeries into five families, establishing a categorization aligned with mainstream LipSync paradigms while accounting for underlying generative mechanisms.

- GAN-based (e.g., Wav2Lip, Talklip): These methods commonly adopt adversarial optimization to minimize discriminator losses, whether operating in landmark or pixel space. As a result of this optimization process, the generated lip-sync videos manifest specific spectral anomalies, particularly in the high-frequency domain.

- Diffusion-based (e.g., Sonic, DreamTalk): Relies on Iterative Denoising. The motion trajectories often exhibit characteristic stochasticity and smoothness derived from the Gaussian sampling process, distinct from the sharp discontinuities of GANs.

- Transformer-based (e.g., IP-LAP): Transformer-based methods treat LipSync as a Sequence-to-Sequence translation task, often leading to specific quantization artifacts or step-wise motion patterns.

- VAE-based (e.g., SadTalker): Relies on Latent Space Distribution. The motion is constrained by the KL-divergence loss, typically resulting in overly smooth, mean-reverting motion trajectories lacking high-frequency micro-expressions.

- Statistical Models (e.g., DiNet): Relies on Geometric Warping. These methods do not synthesize new texture but warp existing pixels, leaving traces of stretching and non-linear distortion without generative noise.

### B.3. Inter-Class Signal Motivation

**Video-Driven Paradigm** The intrinsic coupling between a speaker's unique physiological structure and their kinematic patterns renders identity-motion decoupling an inherently ill-posed problem. When the motion field of a driver is forcibly decoded onto a distinct source identity, it inevitably violates the source's underlying conditional distribution, $P_{src}(\text{Motion}|\text{Identity})$. This incompatibility forces the generator to synthesize frames that fall outside the source's natural manifold, leading to a distributional shift. Consequently, the model struggles to maintain spatial-temporal consistency under these foreign constraints, manifesting as localized temporal artifacts and incoherent micro-expressions that serve as detectable cues for video-driven forgeries.

**Audio-Driven Paradigm**

- **1) Reconstruction-based methods** typically treat LipSync as a localized inpainting task by masking the lower facial region and reconstructing it to maximize the semantic alignment between audio features and synthesized lip patches. The fundamental cause of inconsistency in these methods lies in the asynchrony of data sources. While the lip dynamics $L$ are newly generated from the driving audio $A$, the head pose $H$ is directly inherited from a static reference frame, resulting in a distinct temporal asynchrony where the mouth exhibits speech-driven activity while the head remains rigidly static or executes repetitive movements unrelated to the verbal emphasis

- **2) Diffusion-based methods** leverage denoising diffusion probabilistic models to achieve high-fidelity textures and smoother transitions via iterative denoising from Gaussian noise. However, the stochastic nature of the denoising process introduces high-frequency randomness that tends to wash out the fine-grained synchronization signals between the two modalities. To mitigate jitter, current engineering practices often rigidly anchor the generated head pose to the original poses of the source video frames.

- **3) Independent Learning methods** explicitly decompose the generation task into separate streams for lip and pose features conditioned on the same audio input. This design fails to capture biological coupling because the mapping properties are heterogeneous. The lip mapping is largely deterministic, while the head pose mapping is probabilistic and one-to-many. Training these modalities independently without a shared joint constraint cuts the gradient flow necessary to enforce phase-locking between phoneme articulation and head micro-movements. Consequently, this results in a physiological desynchronization, where the generated lip and pose sequences appear individually plausible but lack the intrinsic conditional dependence found in authentic human speech.

## C. Additional Experimental Setup

### C.1. Evaluation Dataset Selection

We carefully selected three LipSync datasets as follows:

- **AVLips** (Liu et al., 2024) is an audio-visual forgery dataset. It uses the LRS2(Chung et al., 2017) dataset as the audio driving signal and generates forged videos using the Wav2Lip (Prajwal et al., 2020) and MakeITalk (Zhou et al., 2020) models. The dataset comprises 3,396 real and 4,206 forged videos, which we split into training, validation, and test sets following a 7:1.5:1.5 ratio.

- **TalkHeadBench** (Xiong et al., 2025) contains forged videos generated by advanced diffusion models, such as Hallo (Xu et al., 2024a). Since the forged videos in this dataset lack corresponding real videos and some also lack audio, we select 476 real videos from its source, CelebV-HQ (Zhu et al., 2022), and pair them with 448 forged videos that include audio to construct our test set.

- **Celeb-DF** (Li et al., 2020) is a well-known standard DeepFake dataset focusing on identity-swapping forgeries. Its real videos are sourced from celebrity interviews on YouTube. We follow its standard protocol and use its test set of 980 videos for evaluation.

## C.2. Rationale for Baseline Selection

To ensure a comprehensive and fair comparison, we select several SOTA DeepFake detectors as baselines. We assess 8 baselines that rely solely on visual information, covering the following predominant approaches:

- Lip-based: Methods focusing on lip movement anomalies, e.g.,LipForensics (Haliassos et al., 2021), which leverages lip-reading as a proxy task to extract forgery-sensitive features.

- Artifact & Temporal Inconsistency: Methods that capture visual artifacts or temporal discontinuities. This includes FTCN (Zheng et al., 2021) (targeting temporal micro-incoherence), RealForensics (Haliassos et al., 2022) (using temporal self-consistency as an anomaly signal), and TALL (Xu et al., 2024b) (a tamper-aware approach that explicitly localizes manipulated regions).

- Frequency-Domain: Methods assuming forgeries leave detectable traces in the frequency spectrum. This includes CADDM (Dong et al., 2023) (exploiting complementary cues in color-frequency domains), FreqNet (Tan et al., 2024a) (detecting global high-frequency peaks), and NPR (Tan et al., 2024b) (amplifying noise anomalies via Noise Power Residual maps).

We also evaluate 7 multimodal baselines that exploit cross-modal information:

- Discrepancy & Asynchrony: Methods that detect inconsistencies or asynchrony between audio and visual streams. This includes AVAD (Feng et al., 2023) (explicitly detecting A-V asynchrony) and AltFreezing (Wang et al., 2023b) (using an alternating training strategy to enforce A-V synchronization).

- Alignment-based: Methods that explicitly align A-V features to spot forgeries. This includes AVH-align (Smeu et al., 2025) (framing LipSync as a hierarchical cross-modal contrastive learning task), LipFD (Liu et al., 2024) (detecting alignment in faked lip regions) and FGMDC (Yin et al., 2024) (leveraging fine-grained modal-complementary cues from the mouth, teeth, and audio pitch).

- Other Multimodal Approaches: This group includes SpeechForensics (Liang et al., 2024) (detecting acoustic artifacts from TTS/VC in the audio stream), and DFD-FCG (Han et al., 2025) (a frequency-cognition guided A-V framework).

## C.3. Baseline Adjustments for Attribution Task

Our framework includes a model attribution stage, which requires a multi-class classifier to identify the forgery source. We found that most visual-only detectors (e.g. CADDM (Dong et al., 2023), FreqNet (Tan et al., 2024a) ) are ill-suited for this task. Their detection often relies on low-level artifacts that lack sufficient discriminative power between different generative models. Consequently, we selected baselines more amenable to multi-class attribution. This includes TALL (Xu et al., 2024b) and several A-V methods (AVAD (Feng et al., 2023), SpeechForensics (Liang et al., 2024), and AVH-align (Smeu et al., 2025)). For these selected baselines, we adapted their output layers (originally 4-way) to a 5-way classification task (5 distinct forgery source classes) and trained them using a standard cross-entropy loss.

## C.4. Evaluation Protocol of Main Results

**Training and test composition.** LipDA is trained on the LipSync-A training split, which comprises 1,614 authentic videos and 2,312 forged videos produced by 10 generators spanning seven architectural paradigms: Hybrid Transformations (X2Face, 89), Keypoint-Driven (TPSM/LIA/FaceVid, 1,245), Statistical Models (DiNet, 134), GAN-based (MakeItTalk/Wav2Lip, 297), VAE-based (SadTalker, 206), Transformer-based (IP-LAP, 210), and Diffusion-based (DreamTalk, 131). For zero-shot evaluation against unseen SOTA generators, we curate 100 videos from the official OmniSync release, and additionally synthesize 300 videos each for Sonic, KDTalker, and InfiniteTalk using their official pretrained weights, pairing LaPa source identities with VCTK driving audio. Wan-S2V is further evaluated on 85 videos obtained through its commercial API. Corresponding authentic samples are drawn from AVLips to maintain balanced class proportions.

**Baseline reproduction protocol.** To ensure equitable comparison, the 15 baselines are reproduced according to the availability of their training resources. (i) Methods that do not release training code (FTCN, LipForensics, AltFreezing, FGMDC, RealForensics, AVAD) are evaluated using their officially released pretrained weights. (ii) Image-level detectors (NPR, FreqNet, UnivFD) are likewise evaluated with official weights, as their image-only pipelines are incompatible with the video-level inputs required by LipSync forensics. (iii) Video-level detectors whose training code is publicly released (CADDM, TALL, LipFD, AVH-align, DFD-FCG, SpeechForensics) are fine-tuned on the same LipSync-A training split used by our method, guaranteeing that all multimodal comparisons share identical data access. To verify that fine-tuning preserves the baselines' original capability, we additionally re-evaluate each fine-tuned model on its source benchmark. The reproduced accuracies closely track the originally reported ones—e.g., on FF++: CADDM 99.7→94.1, TALL 99.8→99.0, DFD-FCG 99.2→95.1, SpeechForensics 97.6→95.9; on AVLips: LipFD 93.1→96.1, AVH-align 86.3→88.4—confirming the integrity of our reproductions.

**Analysis of frequency-baseline AUC** We further investigate the low AUC reported for NPR (14.4%) and FreqNet (9.4%) on LipSync-A. A frame-level study over 160,532 test frames reveals two compounding effects. First, both detectors classify nearly all inputs as authentic, yielding a false-negative rate of 99.8%. Second, and more critically, real frames consistently receive *higher* forgery scores than forged ones (mean score: real 0.028 vs. fake 0.002), which inverts the ROC ordering and drives AUC below the random baseline. We attribute this counter-intuitive behavior to a domain mismatch between the source training distribution and LipSync forgeries. NPR and FreqNet are pretrained on ForenSynths/CNN-Detection, where forgeries are dominated by the high-frequency upsampling artifacts inherent to GAN-based full-face synthesis. LipSync forgeries, by contrast, modify only the lip region and apply temporal smoothing during reconstruction, which actively *suppresses* high-frequency content. Consequently, the natural micro-textures and rapid articulation of authentic lips elicit stronger frequency-domain responses than the smoothed synthetic counterparts, prompting both detectors to systematically rank real samples as more suspicious. This finding reinforces our central motivation: LipSync forgeries elude the artifact-based assumptions underpinning conventional detectors, and demand a fundamentally different forensic paradigm grounded in physiological coupling.

## D. Additional Robustness Study

To further assess the resilience of our proposed model, we conduct a comprehensive robustness study against a wide array of both visual and audio perturbations. These corruptions are designed to simulate common real-world scenarios, such as post-processing and re-compression.

### D.1. Visual Robustness

We evaluate robustness to visual corruptions by applying seven common perturbations in video processing, including Color Saturation (CS), Contrast (CC), Block-wise occlusion (BW), Gaussian Noise (GN), Gaussian Blur (GB), JPEG compression, and Pixelation (PXL), The specific hyperparameters corresponding to the five intensity fot each perturbation are detailed in Table 8.

*Table 8.* **Robustness experiment parameters**. Each perturbation method employs five unique sets of hyperparameter values, modifying them solely during the video preprocessing phase.

| Type | Parameter | L1 | L2 | L3 | L4 | L5 |
|---|---|---|---|---|---|---|
| Block-wise | block number | 16 | 32 | 48 | 64 | 80 |
| Color Contrast | contrast factor | 0.85 | 0.725 | 0.6 | 0.475 | 0.35 |
| Color Saturation | saturation gain | 0.4 | 0.3 | 0.2 | 0.1 | 0.0 |
| Gaussian Blur | kernel size | 7 | 9 | 13 | 17 | 21 |
| Gaussian Noise | variance $\sigma^2$ | 0.001 | 0.002 | 0.005 | 0.01 | 0.05 |
| JPEG Compression | quality drop | 30 | 32 | 35 | 38 | 40 |
| Pixelation | pixel block | 2 | 3 | 4 | 5 | 6 |

We test all models under five levels of intensity for each perturbation. Fig. 10 provides a qualitative visualization of all seven distortion types at a challenging Level-3 intensity. Table 9 presents the quantitative comparison against SOTA baselines at this fixed Level-3 intensity. Our method (Ours) consistently and significantly outperforms all baselines across every perturbation category. Notably, our model achieves an average AUC of 99.11%, surpassing the next-best baseline (LipForensics) by +13.58%. This demonstrates that our model's learned features are far more resilient to common corruptions. While methods

*Table 9.* **Robustness comparison against Level-3 perturbations**. We report AUC (%) on the AVLips test set. Bold indicates the best performance, and underline marks the second-best.

| Perturbation | FTCN | LipFD | LipForensics | RealForensic | **Ours** |
|---|---|---|---|---|---|
| Block-wise | 68.70 | 96.15 | 93.50 | 62.50 | **99.59** |
| Contrast | 75.56 | 80.77 | 87.00 | 56.25 | **99.27** |
| Saturation | 69.41 | 91.28 | 93.25 | 64.58 | **97.71** |
| Gaussian Blur | 57.77 | 51.79 | 82.00 | 50.00 | **99.54** |
| Gaussian Noise | 54.20 | 51.54 | 71.00 | 45.83 | **98.50** |
| Compression | 64.57 | 90.35 | 80.00 | 58.33 | **99.59** |
| Pixelation | 61.14 | 57.18 | 91.99 | 60.42 | **99.56** |
| **Average** | 64.48 | 74.15 | 85.53 | 56.84 | **99.11** |

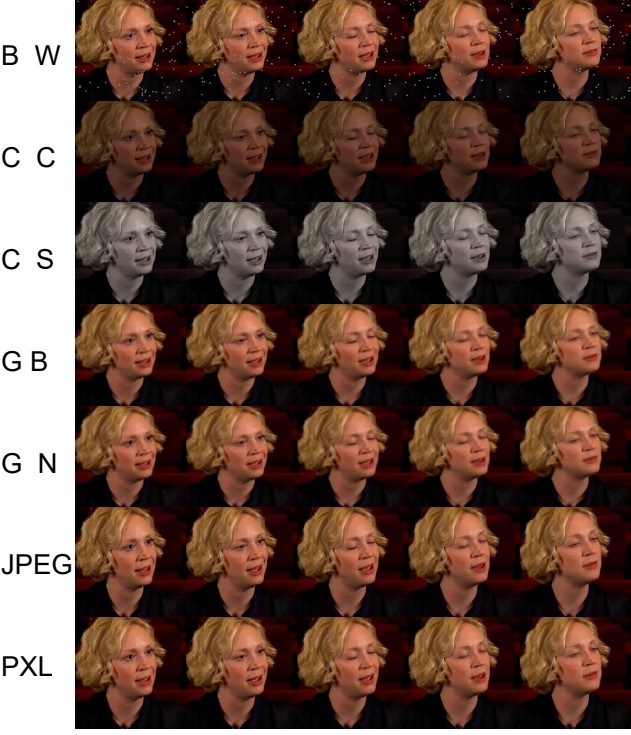

*Figure 10.* **Visualization of the seven perturbation types at Level-3 intensity**.

like FTCN and RealForensic suffer catastrophic performance drops, particularly against noise and blur, our model remains highly stable, with performance (AUC $> 99.5\%$) barely affected by compression, blur, and pixelation, which are ubiquitous in real-world videos.

## D.2. Audio Robustness

We also analyze its resilience to perturbations applied only to the audio stream, and we apply a diverse suite of 12 common audio corruptions, including additive noise, pitch shifting, resampling, room impulse response (RIR) simulation, and time shifting. Table 10 details the performance of our model on the AVLips test set. Our model exhibits remarkable stability against most audio corruptions. Performance remains exceptionally high for perturbations common in real-world scenarios, such as resampling (to 16k or 44k) and reverberation (simulated via RIR). This suggests our model is not overfitting to specific acoustic properties of the training data. The most significant performance drops are observed, as expected, under heavy additive noise (noise_heavy) and significant pitch distortion (pitch_shift_up3, noise_light). Even in these challenging cases, the model maintains a high AUC above $97.3\%$, validating the robustness of our audio-visual fusion mechanism.

*Table 10.* **Performance of our model under various audio corruptions**. Bold and underline denote the best and second-best results per metric (Accuracy, F1-score, AUC), respectively. All metrics are in %.

| Perturbation | Accuracy | F1-score | AUC |
|---|---|---|---|
| noise_heavy | 86.25 | 85.82 | 98.79 |
| noise_light | 70.40 | 63.50 | 97.97 |
| pitch_shift_down3 | 97.11 | 97.36 | 99.33 |
| pitch_shift_up3 | 79.60 | 77.57 | 97.31 |
| resample_16k | 96.94 | 97.23 | 99.37 |
| resample_44k | 97.37 | 97.59 | 99.56 |
| rir_large_room | 97.46 | 97.68 | 99.53 |
| rir_medium_room | 97.64 | 97.85 | 99.54 |
| rir_small_room | **97.81** | **98.00** | 99.55 |
| silence_random | 97.20 | 97.42 | **99.59** |
| time_shift_backward | 97.20 | 97.42 | 99.56 |
| time_shift_forward | 97.20 | 97.45 | 99.52 |

## D.3. Landmark Extraction Robustness

Since LipDA depends on facial landmarks extracted by Google MediaPipe to construct the head-pose representation in Stage I and the keypoint sequence for the TDM in Stage II, the reliability of landmark extraction under degraded visual conditions warrants explicit examination. We therefore stress-test the extractor on 100 videos (19,834 frames in total) under all seven Level-5 visual perturbations defined in Table 8, reporting both the per-frame landmark Failure Rate (FR) and the resulting detection AUC on the AVLips test set.

*Table 11.* Landmark extraction robustness under Level-5 visual perturbations. FR (%) is the per-frame landmark extraction failure rate; AUC (%) is reported on the AVLips test set. Columns are ordered by increasing FR.

| Metric | Baseline | JPEG | BW | CC | PXL | GB | CS | GN |
|---|---|---|---|---|---|---|---|---|
| FR $\downarrow$ | 6.0 | 6.3 | 6.6 | 8.8 | 11.4 | 11.7 | 13.9 | 34.7 |
| AUC $\uparrow$ | 99.72 | 99.60 | 99.69 | 96.99 | 98.55 | 98.33 | 92.10 | 90.98 |

As shown in Table 11, the MediaPipe extractor remains stable under the majority of perturbations: JPEG compression, block-wise occlusion (BW), and color contrast (CC) all keep FR below $9\%$ and AUC above $96\%$. Gaussian noise (GN) constitutes the most adversarial case, elevating FR to $34.7\%$, yet LipDA still attains $90.98\%$ AUC. We attribute this resilience to two compounding factors. First, our sliding-window mechanism discards any window in which more than half of the $T$ frames fail landmark extraction; even under GN-L5, $86.57\%$ of windows remain valid, providing sufficient temporal evidence for reliable inference. Second, the AUC degradation does not strictly track FR: color saturation (CS-L5) exhibits a moderate FR of $13.9\%$ but a sharper AUC drop, because saturation degradation simultaneously compromises the appearance features extracted by the lip encoder rather than the landmark pipeline alone. We further note that perturbations severe enough to substantially impair landmark extraction also render the videos perceptually implausible, thereby diminishing

their deceptive utility in realistic adversarial scenarios.

## E. Extended Discussion

### E.1. Open-set Attribution

The continual emergence of new LipSync architectures renders the universe of potential forgery sources open-ended, raising a legitimate concern about the closed-set assumption underlying our attribution formulation. A direct open-set solution would replace supervised classification with unsupervised clustering over fingerprint features, but such schemes typically suffer from clustering instability and pseudo-label drift, yielding outputs that lack the interpretability required by forensic investigators. We therefore adopt a structured alternative that groups generators by their underlying architectural paradigm following the LipSync taxonomy of Meng et al. (2024), since generators within the same family share fundamental synthesis mechanisms such as iterative denoising or adversarial optimization, and consequently imprint structurally similar temporal fingerprints. This grouping confers an implicit forward compatibility, allowing a newly released generator to be mapped to its corresponding family even when its specific instance was unseen during training. We verify this property on held-out generators, finding that OmniSync and KDTalker are correctly assigned to the Diffusion family at $90.9\%$ and $97.3\%$ accuracy respectively, while EAT (Gan et al., 2023) is attributed to the Transformer family at $99.0\%$. Supervised family-level classification is nevertheless not a complete open-set solution, and future generators built on entirely novel paradigms will require explicit out-of-distribution detection or open-set recognition, extending LipDA with confidence-aware rejection thresholds and incremental family discovery therefore remains a promising direction for future work.

### E.2. Connection to Generative Fingerprint Methods

The notion of generative fingerprints has been extensively studied in image-level forensics, where works such as ManiFPT (Song et al., 2024) and Riemannian-Geometric Fingerprints (Song & Itti, 2025) analyze the statistical signatures left by generative architectures in pixel space. These methods primarily target full-face DeepFakes synthesized by image generators, where forgery cues manifest as spatial frequency artifacts or geometric distortions in a learned latent manifold. LipSync forgeries differ fundamentally in their forgery characteristics, since they preserve the authentic identity and visual context while modifying only localized lip dynamics, leaving spatial fingerprints that are easily diluted by the surrounding real content. As a result, the static pixel-level signatures exploited by prior methods cannot be directly transferred to LipSync attribution. Our framework instead extracts fingerprints from the temporal dynamics of head motion and the cross-modal synchronization between lip articulation and audio, capturing how each generator family modulates the lip-pose coupling across time. To the best of our knowledge, no prior study has systematically investigated LipSync-specific attribution, and our work therefore offers a complementary perspective that extends the fingerprinting paradigm from spatial signatures in synthetic images to temporal signatures in audio-driven facial videos.

## F. Extended Experimental Results

### F.1. Per-Generator Detection on LipSync-A

To provide a fine-grained view of how detection performance varies across generator paradigms, Table 12 reports the per-generator breakdown of the visual-only variant of LipDA on the LipSync-A test split. Each generator's forged samples are scored against the shared pool of 346 authentic samples, with video-level accuracy and AUC jointly reported in the ACC/AUC format. LipDA delivers consistently strong detection across the ten generators covered by the training distribution, attaining an average ACC of $94.6\%$ and an average AUC of $97.4\%$. Detection is essentially saturated on the video-driven generators (X2Face, TPSM, LIA, FaceVid) and on the diffusion-based DreamTalk, while the audio-driven GAN-, Transformer-, and statistical-based generators are reliably identified. The marginally lower AUC observed on SadTalker is attributable to its tendency to inherit near-static head poses directly from the source frame, which compresses the dynamic range of the lip-pose inconsistency signal that our framework exploits. The breakdown therefore confirms that the physiological coupling cue generalizes broadly across diverse synthesis paradigms rather than relying on artifacts specific to any single generator family.

*Table 12.* Per-generator detection performance of the visual-only LipDA on the LipSync-A test split. Each cell reports video-level binary classification accuracy and AUC in ACC/AUC (%) format, both computed against the shared pool of 346 authentic samples. Generators are listed in alphabetical order.

| Generator | DiNet | DreamTalk | FaceVid | IP-LAP | LIA | MakeItTalk | SadTalker | TPSM | Wav2Lip | X2Face | **Avg.** |
|---|---|---|---|---|---|---|---|---|---|---|---|
| **ACC / AUC (%)** | 94.5/98.0 | 95.4/99.9 | 95.5/99.9 | 94.4/97.1 | 95.6/99.9 | 94.8/96.8 | 88.9/84.2 | 96.9/99.9 | 95.0/98.0 | 95.3/100.0 | **94.6/97.4** |

*Table 13.* Fine-grained 15-way model-level attribution performance on the LipSync-A test split. Generators are grouped by their architectural family for ease of comparison. All metrics are reported in %.

| Family | Generator | Prec. | Recall | F1 |
|---|---|---|---|---|
| Hybrid Transformations | X2Face | 72.2 | 86.7 | 78.8 |
| Keypoint-Driven | TPSM | 71.4 | 33.3 | 45.5 |
| | LIA | 71.9 | 76.7 | 74.2 |
| | FaceVid | 46.7 | 46.7 | 46.7 |
| Statistical | DiNet | 96.4 | 90.0 | 93.1 |
| GAN-based | MakeItTalk | 100.0 | 96.7 | 98.3 |
| | Wav2Lip | 80.0 | 66.7 | 72.7 |
| | TalkLip | 65.8 | 83.3 | 73.5 |
| VAE-based | SadTalker | 89.3 | 83.3 | 86.2 |
| Transformer-based | IP-LAP | 86.2 | 83.3 | 84.8 |
| Diffusion-based | DreamTalk | 96.8 | 100.0 | 98.4 |
| | Sonic | 90.6 | 96.7 | 93.6 |
| | KDTalker | 96.8 | 100.0 | 98.4 |
| | OmniSync | 100.0 | 100.0 | 100.0 |
| | InfiniteTalk | 78.4 | 96.7 | 86.6 |
| **Overall (macro avg.)** | | **82.8** | **82.7** | **82.0** |

## F.2. Fine-Grained 15-Way Model-Level Attribution

The main attribution results in Table 3 adopt a 5-way label space defined by architectural family, since family-level granularity offers forward compatibility to unseen generators built on the same paradigm. A complementary question, however, is whether LipDA can also distinguish individual model instances inside the same family, for instance separating Wav2Lip from TalkLip. To probe this, we retrain the Stage II attribution classifier with a 15-way model-level label space that assigns every generator in LipSync-A its own class, holding all other components, hyperparameters, and training data identical to the original recipe. Because the supervised target is fundamentally different from the 5-way protocol, the two sets of numbers are not directly comparable, and the 15-way results are reported here as a complementary stress test rather than as a refinement of the family-level results in the main paper.

Under this finer-grained protocol, LipDA attains an overall accuracy of $82.67\%$ and a macro-average F1 of $82.04\%$ on the LipSync-A test split. Table 13 reports the per-model breakdown grouped by family. Three observations emerge. First, the diffusion family remains highly separable at the model level, with DreamTalk, KDTalker, and OmniSync all attaining $100\%$ recall and Sonic reaching $96.7\%$, indicating that distinct diffusion instances retain individually distinguishable temporal fingerprints despite their shared denoising backbones. Second, GAN-based instances are also distinguishable, with MakeItTalk, Wav2Lip, and TalkLip reaching recalls of $96.7\%$, $66.7\%$, and $83.3\%$ respectively, confirming that within-family separation is achievable for the GAN paradigm. Third, the most pronounced residual confusion concentrates inside the keypoint-driven family, where TPSM, LIA, and FaceVid share landmark-based motion priors and consequently produce highly similar temporal patterns, leading to mutual misclassification. This within-family confusion is consistent with the rationale behind our family-level taxonomy and supports the family-level grouping adopted in the main paper as a more reliable operating point for downstream forensic deployment.

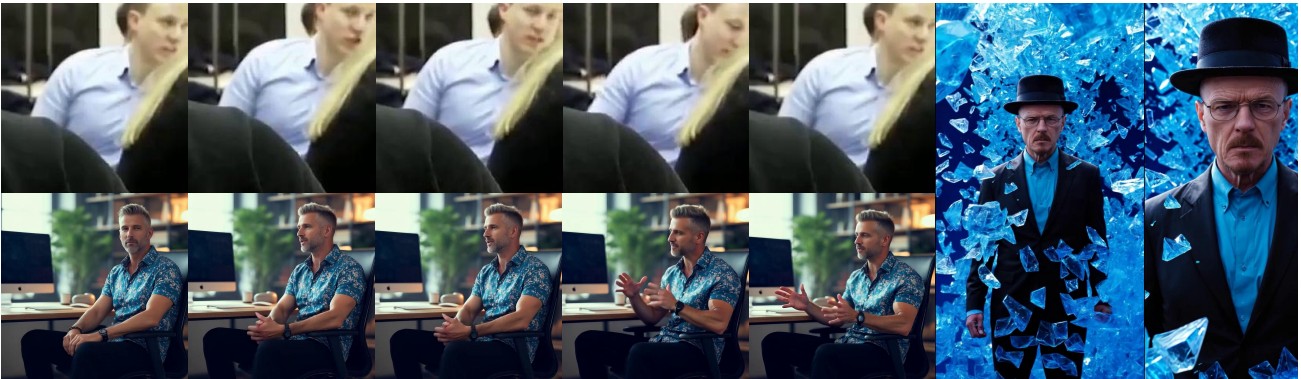

*Figure 11.* Representative failure cases of LipDA on the LipSync-A test split. All three patterns are also failed by the baselines evaluated in Table 2, indicating a shared operational limit rather than a weakness specific to our framework.

### F.3. Generalization to Full-Video Generation

The recent emergence of full-video generation paradigms, which synthesize the entire frame rather than locally editing the lip region, introduces a new failure surface for forgery detectors that rely on localized visual artifacts. To assess whether LipDA's physiological coupling signal remains discriminative in this regime, we conduct an extended zero-shot evaluation that incorporates the commercial-grade full-video generator Wan-S2V (Wan et al., 2025) alongside four recent diffusion-based generators excluded from training. As reported in Table 14, LipDA achieves an average detection accuracy of $87.0\%$ across the five unseen generators, with $88.2\%$ on Wan-S2V specifically, indicating that the lip-pose coupling cue transfers naturally to forgeries where the entire video is synthesized rather than locally edited.

*Table 14.* Zero-shot detection accuracy (%) of LipDA on unseen SOTA generators, including the full-video generator Wan-S2V. All models are evaluated without any fine-tuning on LipSync-A.

| **Generator** | OmniSync | Wan-S2V | Sonic | KDTalker | InfiniteTalk |
|---|---|---|---|---|---|
| **ACC (%)** | 95.5 | 88.2 | 87.8 | 84.6 | 78.8 |

We further verify the Stage II attribution framework on representative recent generators under the same zero-shot protocol, with 300 samples generated per model. Specifically, LatentSync (Li et al., 2024) is correctly attributed to the Diffusion family at $97.5\%$ accuracy, consistent with $97.3\%$ on KDTalker and $90.9\%$ on OmniSync. Together, these detection and attribution results indicate that both the physiological inconsistency cue and the temporal motion fingerprint exploited by LipDA reflect general properties of audio-driven facial synthesis rather than artifacts specific to the LipSync editing pipelines on which our framework was trained.

### F.4. Bad Case Analysis

To better characterize the operational boundary of LipDA, we examine its failure cases on the LipSync-A test split and identify three recurring patterns, all of which are shared by every evaluated baseline rather than being unique to our framework. Figure 11 visualizes representative examples.

- **Stationary speaker under complex camera motion**, where the head remains nearly motionless relative to the body and the lip-pose inconsistency signal that our framework relies on becomes inherently weak.

- **Persistent downward gaze**, which renders the lip region largely invisible across frames, depriving the lip encoder of effective input and collapsing the contrastive comparison in Stage I.

- **Prolonged facial occlusion** by hands, microphones, or hair, which destabilizes landmark extraction and propagates noisy pose features into the temporal modules.

These cases collectively delineate a shared operational limit of detection paradigms that depend on visible lip dynamics and well-defined head motion as primary forensic cues, motivating future work on confidence-aware inference and complementary signals drawn from upper-face dynamics or full-body movement.

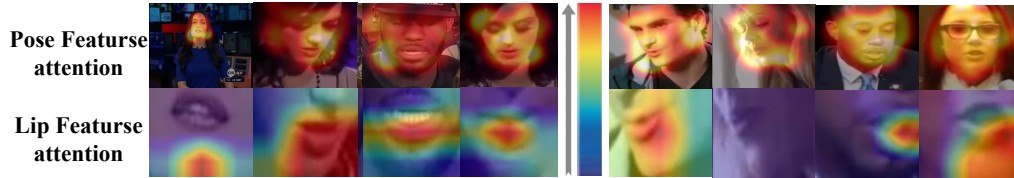

*Figure 12.* **Spatial attention maps for the dual-branch encoders**. Brighter colors indicate higher attentional weight.

### F.5. Attribution Ablation

We ablate our Stage II attribution components in Table 15. The MSM variant achieves a strong 92.92% ACC and 93.64% F1, establishing that audio-visual synchronization patterns are the core discriminative signal for attribution. Conversely, relying only on the TDM features causes a catastrophic performance drop to 74.78% ACC, proving TDM is insufficient alone. However, our Full Model, which combines both MSM and TDM, achieves the highest performance (97.50% ACC). This significant gain confirms that the TDM provides critical complementary temporal cues, validating the synergistic design of our framework.

*Table 15.* **Ablation study on attribution model components**. We report the average Accuracy (ACC, %) and F1-Score (F1, %) across **five generator families** on the LipSync-A.

| Model Variant | ACC | F1 |
|---|---|---|
| Full Model | **97.50%** | **93.90%** |
| temporal_only(TDM) | 74.78% | 73.58% |
| av_sync_only(MSM) | 92.92% | 93.64% |
| audio_only | 90.71% | 91.36% |

### F.6. Dataset Case Analysis

Attention maps in Fig. 12 further reveal a clear functional disentanglement. The pose encoder attends to a broad facial context and the lip encoder is highly localized on fine-grained lip motion. This confirms our core mechanism operates by detecting a mismatch between this global context and local motion, validating our lip-head pose inconsistency hypothesis.

To provide a qualitative overview of the LipSync-A dataset and the distinctive characteristics of different generative families, we present visual comparisons in Fig. 13 and Fig. 14. These visualizations underscore the diversity of forgeries in our benchmark and highlight the model-specific patterns—such as variations in lip articulation sharpness, motion amplitude, and head pose dynamics—that form the basis for our attribution task. The discernible differences across generators validate the premise that each model family imparts a unique fingerprint, which our method is designed to capture.

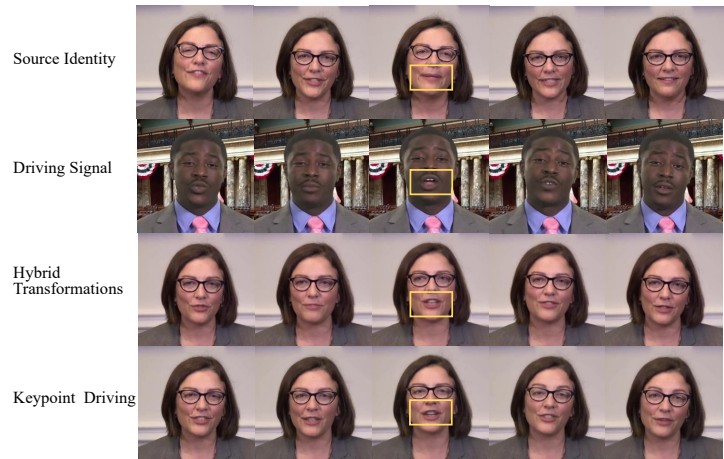

*Figure 13.* **Comparison of forgeries generated from the same source identity and driving video.**

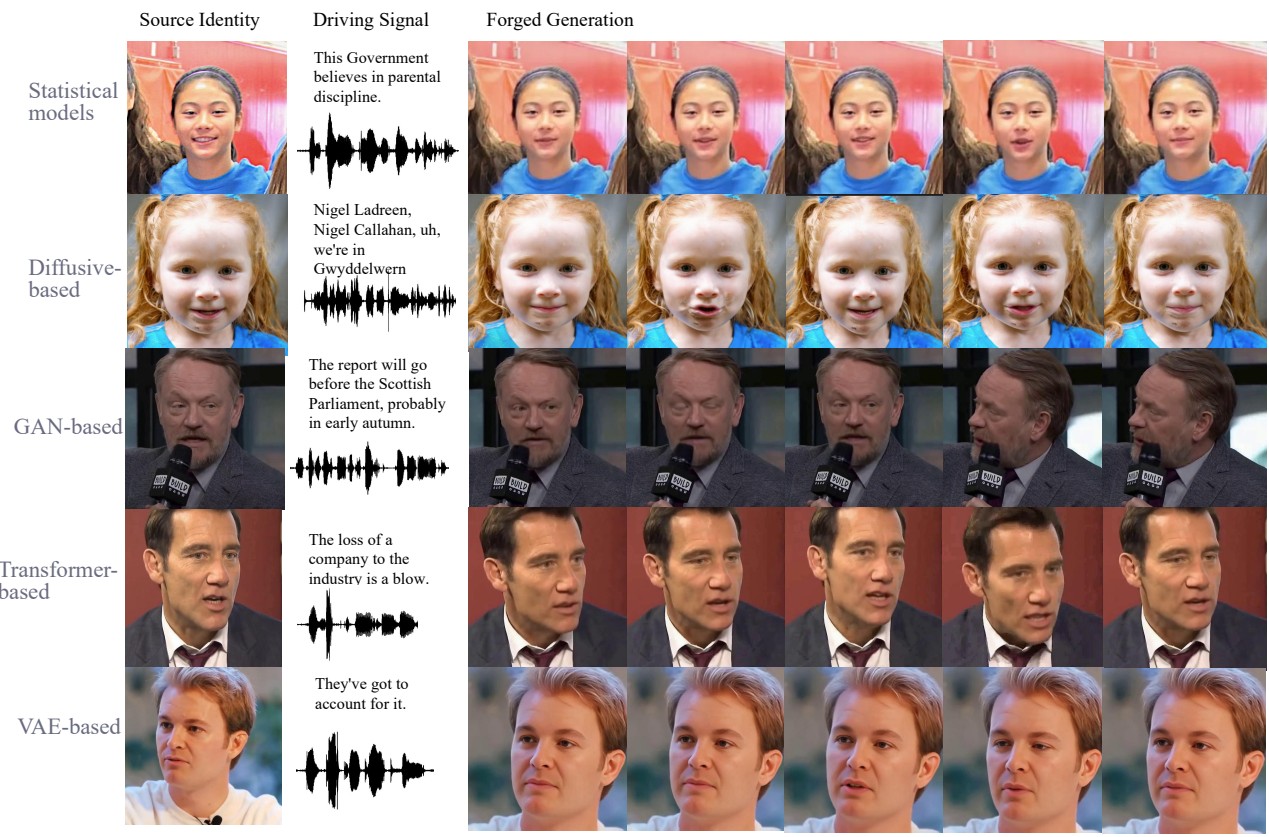

*Figure 14.* **Samples from each of the five audio-driven generator families in LipSync-A.**

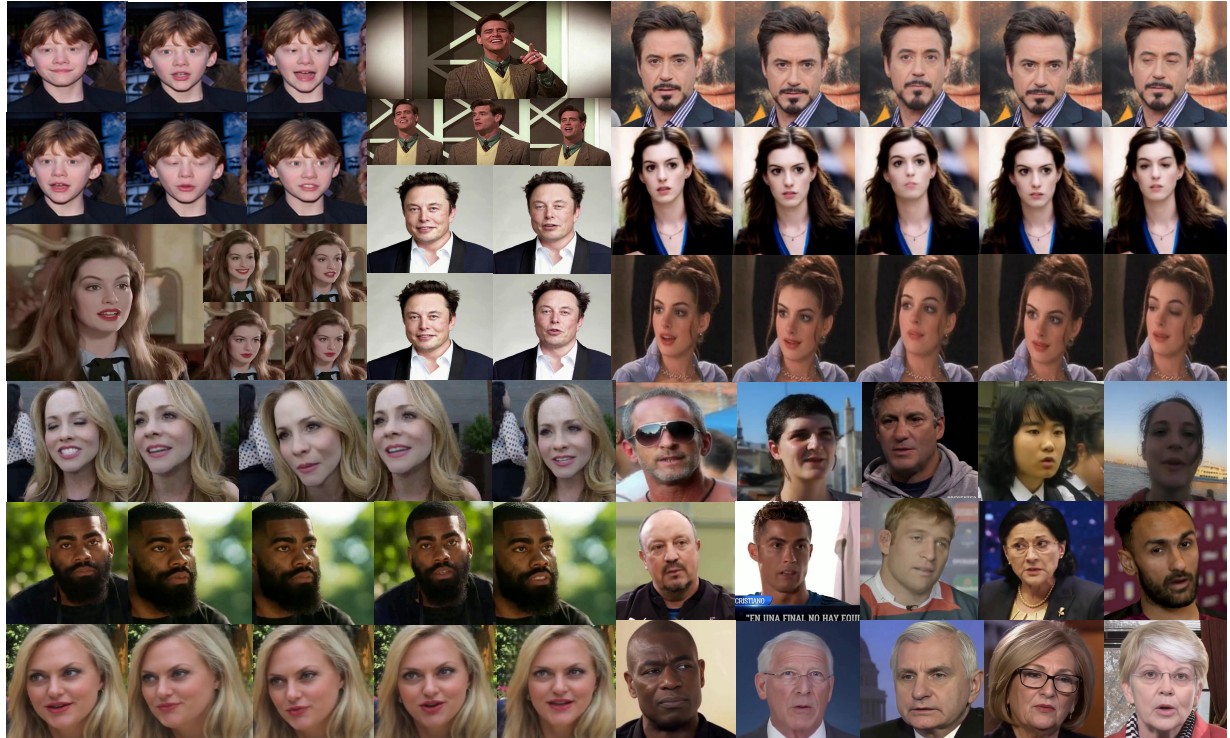

*Figure 15.* **Visualization of samples from our constructed LipSync-A dataset.**

