# OpenReview forum: "Ariadne's Thread of LipSync: Unraveling Forgeries via Inconsistency between Lip Motions and Head Poses"
_ICML.cc/2026/Conference — ICML 2026 regular_

### Official Review · Reviewer_Zqex · 2026-02-20

**Soundness:** 3
**Presentation:** 2
**Significance:** 3
**Originality:** 3
**Overall Recommendation:** 4
**Confidence:** 4

**Summary:**

The paper introduces LipDA, a joint framework for LipSync detection and attribution. The core innovation lies in leveraging the inconsistency between head poses and lip motions to identify and trace the source of forgeries. The authors also contribute LipSync-A, a dataset with fine-grained labels tailored for the attribution task.

**Compliance With Llm Reviewing Policy:**

Affirmed.

**Final Justification:**

Concerns well-solved, I will maintain my current score.

**Key Questions For Authors:**

- Please clarify the masking strategy mentioned in Section 3.1. Was it indeed the lower face that was masked/manipulated?
- Could the authors provide a more detailed explanation (either in the text or by updating Figure 2) of how the audio features interact with the Head-Pose module?
- I encourage the authors to include results or a discussion regarding newer diffusion-based LipSync models (e.g., LatentSync) to demonstrate the generalizability of the attribution framework.
- Please optimize the document's LaTeX layout to eliminate the large gaps below the figures.

**Limitations:**

See the above part.

**Strengths And Weaknesses:**

### Strengths
- Utilizing the physiological and behavioral inconsistency between head movement and speech (lip motion) is a novel and intuitive approach to both detection and attribution.
- The analysis regarding Intra-Class Signals is insightful and provides a strong justification for the proposed feature extraction strategy.
- As shown in Table 2, the proposed method substantially outperforms current SOTA baselines in both detection accuracy and source tracing.
- The experimental section is comprehensive, providing thorough ablation studies and multi-dimensional evaluations that validate the effectiveness of the LipDA framework.

### Weaknesses
- There is a notable contradiction in the technical description. In Section 3.1 (second paragraph), the authors state that they "synthesize lip movements by masking the upper face," which appears to be a typo. In contrast, Section 4.2 correctly notes that LipSync manipulations primarily affect the lower facial region. Such inconsistencies in describing the core synthesis process should be rectified to avoid confusing the reader.
- The current visualization of the pipeline is somewhat opaque. For instance, the process of feeding the audio signal directly into the Head-Pose module is not intuitively explained within the figure.
- There is excessive white space around Figure 2 and Figure 3, which disrupts the flow of the text and impacts the overall professionalism of the manuscript's layout.
- The scope of the attribution task lacks some recent SOTA models. Specifically, among diffusion-based methods, prominent open-source works such as LatentSync are missing from the comparison. Including these would strengthen the claims regarding the method's robustness against modern generative techniques.

---

> ### Author Rebuttal · Authors · 2026-03-31
>
> Dear Reviewer #Zqex, we sincerely thank you for your precious time and valuable comments. We are deeply encouraged by your positive recognition of our **novel and intuitive approach, insightful analysis, and extensive experiments**. We sincerely hope the following clarifications and new experiments can address your concerns.
>
> ___
>
> **W1 & Q1.**  Inconsistent description of the masking strategy in Section 3.1.
>
> **A1:** We sincerely appreciate your careful reading and pointing out this typo. The correct description is “masking the lower facial region,” consistent with Appendix B.3. We will correct this error in the revised manuscript.
>
> ___
>
> **W2 & Q2.** Clarification on how audio features interact with the Head-Pose module in Figure 2.
>
> **A2:** Thank you so much for this helpful feedback! We acknowledge that Fig. 2 does not clearly illustrate how head pose is derived in audio-driven generation.
>
> - We take Sonic (CVPR 2025) as a representative example to clarify. Sonic decomposes talking face generation into two independent sub-problems “audio-to-lip” and “audio-to-head-pose“. For head pose, the raw audio is first processed by a pretrained Whisper-Tiny encoder to extract multi-scale audio features. These audio features and CLIP-encoded reference image embedding are fed into a 3-layer MLP that learns the audio-to-head-motion mapping from massive training data, producing two scalar parameters for head translation amplitude and expression strength. During inference, these two parameters are injected into the diffusion U-Net's ResNet blocks via position encoding, influencing the denoising process to control the head pose.
> - We will update Fig. 2 to explicitly illustrate the audio encoding step, the dedicated module for learning head pose parameters from audio, and the injection of these parameters into the diffusion process for head motion control, clearly conveying the decoupled generation mechanism.
>
> ___
>
> **W3 & Q4**.  Optimization of the LaTeX layout to reduce white space
>
> **A3:** We thank you for this suggestion. We will optimize the LaTeX layout by adjusting figure placement and reducing excessive white space around Figures 2 and 3 to improve readability and presentation quality.
>
> ___
>
> **W4 & Q3**.  Evaluation of the attribution framework on representative SOTA LipSync models, e.g., LatentSync.
>
> **A4:** Thank you for your broad knowledge of the latest LipSync generation landscape and for bringing LatentSync [6] to our attention. Following your suggestion, we conducted supplementary evaluation with LatentSync using official v1.6 weights and 300 generated samples under a strict zero-shot protocol. Our method achieves **97.5%** attribution accuracy on LatentSync, consistent with **97.3%** on KDTalker and **90.9%** on OmniSync, demonstrating **strong generalizability of our attribution framework to unseen modern generative techniques**. We will include these results and a discussion of LatentSync in our revised manuscript. Thank you again for your recognition and this valuable feedback!
>
> [6] Latentsync: Taming audio-conditioned latent diffusion models for lip sync with syncnet supervision. arViv2024.

---

> > ### Author Rebuttal · Reviewer_Zqex · 2026-04-03
> >
> > Thank you for the supplementary evaluation on LatentSync. The 97.5% accuracy is encouraging, but a single number alone is not sufficient — I would expect these results to be integrated into the existing evaluation tables with the same metrics and protocol, including per-generator breakdowns.
> >
> > Additionally, recent representative LipSync models (LatentSync, etc.) are directly relevant threat models for this work. I strongly encourage the authors to properly cite and discuss them in the related work and experiment sections.
> >
> > I will maintain my current score.

---

> > > ### Author Response · Authors · 2026-04-03
> > >
> > > Dear reviewer, thank you so much for your positive feedback! It encourages us a lot! We will ensure that the revised version includes the new experiments and corresponding discussions, including per-generator breakdowns. We also agree that LatentSync and other recent representative LipSync models are directly relevant threat models. We will properly cite and discuss them in both the related work and experiment sections in our revision. We believe this will help readers better understand the threat landscape that our detection method is designed to address.

---

### Official Review · Reviewer_VXw8 · 2026-03-12

**Soundness:** 3
**Presentation:** 3
**Significance:** 3
**Originality:** 3
**Overall Recommendation:** 4
**Confidence:** 3

**Summary:**

The paper presents LipDA, a two-stage framework for jointly detecting LipSync forgeries and attributing them to their source model. The central insight is that natural speech produces biologically coupled lip and head movements, whereas current generators decouple these by driving lips from audio while sampling head pose independently. Stage 1 leverages this through a dual-branch contrastive learning setup that aligns real lip-pose pairs and separates forged ones, and Stage 2 builds on frozen Stage I encoders with temporal dynamics and audio-visual synchronization modules to classify fakes into five generator families. The work also contributes LipSync-A, a reasonably large dataset of 16,000 videos from 15 generators across 7 architectures. Reported results are strong, with over 97% detection AUC and 97.5% attribution accuracy, along with promising cross-domain generalization to existing benchmarks.

**Compliance With Llm Reviewing Policy:**

Affirmed.

**Key Questions For Authors:**

1. Can the authors report 15-way model-level attribution accuracy instead of 5-way family-level? If two GAN-based forgeries (e.g., Wav2Lip vs. TalkLip) appear in the wild, can LipDA distinguish them?

2. The Stage II attribution classifier is trained only on forged samples. What happens when a real video is passed to the attribution branch during inference? Does the pipeline always gate on the Stage I detection output, or can the attribution classifier produce a (meaningless) output for real videos?

3. How robust is the landmark extraction step under the visual perturbations tested? What fraction of frames fail landmark detection under Level-5 Gaussian blur or noise?

**Limitations:**

1. The framework relies on extracting 468 facial landmarks and computing 6-DoF head pose vectors. But landmark extraction has known failure modes: low resolution, extreme poses, occlusions, compression artifacts. The paper doesn't discuss how landmark extraction errors propagate into detection performance. If the landmark detector itself struggles with compressed or noisy video (precisely the robustness conditions tested in Section 5.4), the pipeline could fail silently. The robustness results in Figure 6 are encouraging, but it would be informative to know what fraction of frames have landmark extraction failures under each perturbation level, and how the method handles missing or noisy landmarks.

2.  The paper reports aggregate metrics but never shows failure cases. Where does LipDA produce false positives (real videos classified as fake)? Where does it miss forgeries (fake classified as real)? Are certain generator types harder to detect? Table 2 reports aggregate AUC but it would be nice to see per-generator detection breakdown.

3. The paper seems to have missed a relevant work by Yang et al. [1]. Can the authors distinguish the main differences from this existing work? The paper will be left with ambiguity and incomplete comparison with the previous work without citing and distinguishing with this paper.

**References**
[1] Yang, Xin, Yuezun Li, and Siwei Lyu. "Exposing deep fakes using inconsistent head poses." ICASSP 2019-2019 IEEE international conference on acoustics, speech and signal processing (ICASSP). IEEE, 2019.

**Strengths And Weaknesses:**

1. The paper's main argument, which considers LipSync generators disrupt the intrinsic coordination between lip motion and head pose because they model these as independent sub problems is both physiologically motivated and verified through the proposed architecture.

2. The AU analysis intensity results clearly shows concrete empirical validation of the core hypothesis.

3. Stage 1 and Stage 2 of the architecture addresses genuinely different tasks with appropriate inductive biases. The model architecture is clean and each component of the model is well-ablated.

---

> ### Author Rebuttal · Authors · 2026-03-31
>
> Dear Reviewer #VXw8, we sincerely thank you for your precious time and valuable comments. We are deeply encouraged by your positive recognition of our **main argument**, **grounded hypothesis** and **clean architecture design**. We sincerely hope the following clarifications and new experiments can address your concerns.
>
> ___
>
> **K1:** Model-level Attribution
>
> **A1:** Thanks for this constructive suggestion! **Following your valuable comment, we replaced the 5-way family classifier with a 15-way model-level classifier** and kept all other settings identical. **The overall accuracy reaches 82.67% with macro-average F1 of 82.04%**. Diffusion-family models are highly separable, with DreamTalk, KDTalker, and OmniSync all at 100% recall and Sonic at 96.7%. For the reviewer's specific question, GAN-family models can also be effectively distinguished, MakeItTalk, Wav2Lip, and TalkLip achieve 96.7%, 66.7%, and 83.3% recall respectively. We will include the complete 15-way confusion matrix and detailed per-model analysis in our revision.
>
> ___
>
> **K2:** Attribution Behavior on Real Videos
>
> **A2:** We appreciate this practical question. **We clarify that our pipeline strictly gates on the Stage I detection output** during inference. Only videos classified as forged by Stage I are passed to the Stage II attribution classifier. **In false positive cases, the attribution classifier produces low confidence across all generator families** since real videos lack generator-specific fingerprints. **The classifier then labels such samples as "others"**. Furthermore, Stage I achieves over 97% AUC, making such false positive cases rare in practice.
>
> ___
>
> **K3 & L1:** Landmark Extraction Robustness
>
> **A3:** Thank you so much for this helpful feedback! **Following your valuable suggestion, we conducted supplementary experiments on 100 videos** with 19,834 frames under all Level-5 perturbations.
>
> | Condition    | Baseline | JPEG  | BW    | PXL   | GB        | CS    | CC    | GN        |
> | ------------ | -------- | ----- | ----- | ----- | --------- | ----- | ----- | --------- |
> | Failure Rate | 6.0%     | 6.3%  | 6.6%  | 11.4% | 11.7%     | 13.9% | 8.8%  | 34.7%     |
> | AUC          | 99.72    | 99.60 | 99.69 | 98.55 | **98.33** | 92.10 | 96.99 | **90.98** |
>
> LipDA uses Google MediaPipe, a widely adopted and robust landmark library. Results show that it handles most perturbations well. Notably, CS-L5 shows a larger AUC drop despite moderate failure rate, as color saturation degradation additionally impacts the lip encoder's feature extraction. GN-L5 is the most challenging type, yet LipDA still achieves **90.98%** AUC. Our sliding window mechanism excludes windows where over half of T=7 frames fail extraction, and **even under GN-L5, 86.57% of windows remain usable**, providing sufficient temporal evidence for reliable predictions. Furthermore, we note that under extreme perturbations, the visual quality of videos is also severely degraded, largely diminishing their deceptive threat. We'll add the above results and discussions in revision, which we believe would undoubtedly strengthen this work. Sincerely thank you again for your truly inspiring feedback!
>
> ___
>
> **L2:** Per-generator Detection Breakdown
>
> **A4:** Thank you for this constructive suggestion! **Following your valuable comment, we conducted a detailed per-generator detection breakdown and failure analysis** on the test set. All video-driven methods achieve **100%** detection, and among audio-driven methods, DreamTalk also reaches **100%**, while DINet and MakeItTalk achieve **91.9%** and **92.5%** respectively. The most challenging are Wav2Lip at **88.2%** and IP_LAP at **76.7%**, primarily due to lower source identity quality in their test samples. For false positives, these mainly occur when prolonged facial occlusion causes unstable landmark extraction. For false negatives, these concentrate in scenarios with complex camera motion but a stationary speaker, where minimal head movement produces weak lip-head inconsistency signals. We will include complete per-generator results and failure case visualizations in our revision.
>
> ___
>
> **L3:** Comparison with Yang et al.
>
> **A5:** Thanks for your thoroughness, we will cite and discuss this work in our revision. Yang et al. detect face-swapping videos via static per-frame pose errors classified by a simple SVM. Our work reveals and exploits the disrupted consistency between lip motion and head pose in LipSync forgeries via temporal contrastive learning, integrates audio modality through cross-modal attention for synchronization analysis, and unifies detection with source attribution within a single two-stage framework.

---

> > ### Author Rebuttal · Reviewer_VXw8 · 2026-04-03
> >
> > My concerns are fully resolved.

---

> > > ### Author Response · Authors · 2026-04-03
> > >
> > > Thank you again for your positive feedback and your thoughtful review. Your support means a great deal to us and motivates us to continue advancing in the LipSync field.

---

### Official Review · Reviewer_3Ejv · 2026-03-13

**Soundness:** 3
**Presentation:** 3
**Significance:** 2
**Originality:** 3
**Overall Recommendation:** 4
**Confidence:** 3

**Summary:**

This work addresses how to effectively detect and attribute the continuously evolving, high-fidelity LipSync forgery videos. Existing defense strategies primarily rely on local artifacts or explicit audio-visual mismatches, clues that are increasingly failing against advanced generative models. To solve this pain point, the paper proposes a novel two-stage framework called LipDA (joint LipSync Detection and Attribution). Its core idea is based on the inherent biological coupling between lip movements and head poses in natural speech; LipSync algorithms inevitably disrupt this physiological consistency due to their localized masking or decoupled generation mechanisms. In the first stage, LipDA utilizes contrastive learning to align authentic lip and pose features to differentiate between real and fake videos. In the second stage, it combines a Modality Synchronization Module (MSM) and a Temporal Dynamic Module (TDM) to extract the model "fingerprints" of specific generator families to achieve source attribution. Furthermore, the paper constructs LipSync-A, a large-scale attribution dataset encompassing 7 architectures, 15 generators, and a total of 16,000 videos.

**Compliance With Llm Reviewing Policy:**

Affirmed.

**Final Justification:**

I keep my score at "Weakly Accept".

**Key Questions For Authors:**

1.Please explain the anomalous performance of baseline models like FreqNet (AUC: 9.40%) and NPR (AUC: 14.40%) in Table 2, which fall significantly below the 50% random guessing probability.

2.Could you further clarify how the representation $P \in \mathbb{R}^{468 \times 3}$ used for contrastive learning in Stage I circumvents interference from facial micro-expressions? How does the framework guarantee that the network concentrates solely on the "global head pose" rather than simply detecting local facial motion anomalies?

3.Regarding the statement that "all baseline models with officially released training code are fine-tuned under the same setting," do these reproduced baseline performances achieve levels comparable to the numbers reported in their respective original papers?

**Limitations:**

Yes

**Strengths And Weaknesses:**

Strengths

1.The shift in perspective demonstrates a certain degree of foresight and innovation. The authors shift the focus of detection from traditional local texture artifacts or simple audio-visual asynchrony to high-dimensional, global physiological consistency (the biological coupling between lip movements and head poses). This fundamentally captures the inherent structural flaws of current data-driven generative models that utilize a decoupled control paradigm (processing audio-to-lip and audio-to-pose mappings separately).

2.The experimental design is thorough. The paper conducts a comprehensive evaluation across in-domain, cross-domain (AVLips, TalkHeadBench), and unseen generative algorithms (Sonic, KDTalker, etc.), as well as under multimodal perturbations (particularly visual blur, compression, and complex audio noise).

3.The paper is well-written, and the motivation is clearly articulated. The Action Unit (AU) intensity analysis in Figure 3 and the t-SNE visualization in Figure 4 strongly support the authors' core hypothesis: different generative paradigms disrupt the inherent physiological coordination and leave behind separable model fingerprints.

4.The authors are the first to explore Detection and model Attribution within a single unified framework. Furthermore, they construct LipSync-A, the most comprehensive fine-grained LipSync attribution benchmark dataset to date, bridging the gap in high-quality forensic resources in this field and making a solid contribution to the community.

Weaknessess

1.In the Stage I detection methodology, head motion is represented by a sequence of 468 facial keypoints. However, such high-density facial landmarks inherently capture a vast amount of local muscle micro-expressions, rather than purely representing global, rigid head rotation and translation.

2.In the contrastive optimization objective of Stage I, the model enforces the minimization of the $L_2$ distance between pose and lip embeddings for real samples. In a natural state, the relationship between head pose and lip movement is a complex, non-linear dynamic correlation, rather than strictly aligned entities that justify a direct mean squared error calculation in a shared dimensional space. Forcing their projection into the same latent space and applying $L_2$ minimization currently lacks a robust physiological and mathematical justification.

3.Although the paper covers common LipSync detection baselines, comparisons within the specific domain of "generator fingerprinting/attribution" remain insufficient. The current attribution evaluation primarily relies on family-level classification comparisons against several multimodal detectors, rather than systematic benchmarking against specialized source attribution or fingerprinting methods. Consequently, the claims of presenting the "first unified framework" and demonstrating "source tracing superiority" are not entirely convincing.

---

> ### Author Rebuttal · Authors · 2026-03-31
>
> Dear Reviewer #3Ejv, we sincerely thank you for your valuable time and constructive comments! We are encouraged by your recognition of our **clear writing**, **motivation**, and **novelty**. We sincerely hope the following clarifications and new experiments can address your concerns.
>
> ___
>
> **W1 & Q2:** Pose Representation and Micro-expression Concerns
>
> **A1:** We acknowledge that the 'head pose' does not fully reflect the scope of our representation, which we will revise accordingly.
>
> - In practice, our dense 3D landmark temporal representation inherently encodes both global head motion and local facial dynamics. In authentic speech, lip articulation is coordinated with surrounding facial muscles, but LipSync generators synthesize lip pixels directly from audio and disrupt this coupling, as validated in Fig. 3. **Therefore, facial micro-expressions serve as an enhancing discriminative signal for detection rather than interference**.
> - **Our contrastive framework learns holistic coupling, specifically the inconsistency between global pose and local lip motion**. As shown in Fig. 11, the pose encoder attends to broad facial regions while the lip encoder focuses on fine-grained mouth details, confirming that the two branches capture representations at different levels.
>
> ___
>
> **W2:** Contrastive Optimization
>
> **A2:** We appreciate this thoughtful concern. Indeed, lip-head dynamics involve complex non-linear coupling, which is precisely why both features are first transformed by learnable non-linear projection heads with sufficient capacity to encode such correlations into the embeddings. **The goal of Stage I is not to reconstruct this relationship precisely but to learn a discriminative metric space.** This aligns with established practice SyncNet [3], which employs the same $L_2$ paradigm in a shared space to measure audio-visual synchrony. The t-SNE visualization in Fig. 8 directly validates the effectiveness of this learned space despite the simplicity of the $L_2$ objective.
>
> [3] Out of time: automated lip sync in the wild. ACCV2026.
>
> ___
>
> **W3:** Comparison with Fingerprint Methods
>
> **A3:** We respectfully note that existing fingerprinting and attribution methods (e.g., ManiFPT [4], R-GMfpts [5]) primarily target image-level DeepFakes and cannot be directly applied to LipSync video attribution due to differences in forgery characteristics. To our knowledge, **no prior work has systematically studied LipSync attribution, which is the gap our work aims to fill.** We sincerely thank you for highlighting these methods and we will include a detailed discussion of the broader fingerprinting literature in our revision.
>
> [4] ManiFPT: Defining and Analyzing Fingerprints of Generative Models. CVPR2024.
>
> [5] Riemannian-Geometric Fingerprints of Generative Models. ICCV2025.
>
> ___
>
> **Q1:** AUC of FreqNet and NPR
>
> **A4:** Thanks for this careful observation. We evaluated both models using official code and weights with standard sklearn roc_auc_score.
> - **Inspired by your insightful query, we conducted a detailed experimental on 160,532 frames**. We find that both models classify nearly all samples as real with high confidence (**FNR 99.8%**), yet real frames (**mean: 0.0284**) consistently receive higher forgery scores than fake frames (**mean: 0.0021**), directly causing AUC below 50%.
> - We hypothesize that this is **caused by a fundamental domain mismatch** leading to overfitting to the real label. Both baselines are trained on GAN-based full-face syntheses with prominent high-frequency upsampling artifacts, which are largely absent in LipSync forgeries. Moreover, LipSync generators reconstruct lip regions with smoother textures, producing less high-frequency content than authentic lips, causing real samples to score systematically higher. This finding further validates our motivation that LipSync forgeries necessitate a different detection paradigm. We have incorporated this detailed analysis into our revision to better explain the limitations of these baselines.
>
> ___
>
> **Q3:** Fairness of Baseline Reproduction
>
> **A5:** **Following your valuable suggestion, we verified that our fine-tuning preserves baseline capability** by evaluating each method on its originally reported evaluation set. Our 15 baselines fall into three groups: (1) methods without released training code (FTCN, LipForensics, AltFreezing, FGMDC, RealForensics, AVAD), evaluated using official pretrained weights; (2) image-level detectors (NPR, FreqNet, UnivFD), whose training pipelines are incompatible with video-level LipSync data, evaluated using official weights; (3) video-level detectors with released code (CADDM, TALL, LipFD, AVH-align, DFD-FCG, SpeechForensics), fine-tuned on the same LipSync training data as ours to ensure equal data access. **On FF++: CADDM 99.7→94.1, TALL 99.8→99.0, DFD-FCG 99.2→95.1, SpeechForensics 97.6→95.9. On AVLips: LipFD 93.1→96.1, AVH-align 86.3→88.4.** We will include this reproduction analysis in our revision.

---

> > ### Author Rebuttal · Reviewer_3Ejv · 2026-04-03
> >
> > I have no more concerns, and maintain positive rating.

---

> > > ### Author Response · Authors · 2026-04-03
> > >
> > > We are pleased that your concerns have been addressed. We would like to express our heartfelt gratitude for your insightful suggestions and kind support!

---

### Official Review · Reviewer_JKgT · 2026-03-13

**Soundness:** 2
**Presentation:** 3
**Significance:** 3
**Originality:** 2
**Overall Recommendation:** 4
**Confidence:** 4

**Summary:**

This paper addresses the detection of LipSync-generated videos, which have become increasingly realistic and difficult to identify. The authors propose LipDA, a framework for joint LipSync detection and source attribution, based on the observation that natural speech videos exhibit an inherent coupling between lip movements and head poses. LipDA detects forgeries by measuring inconsistencies between lip and pose features, and performs attribution by capturing model-specific temporal dynamics and audio–visual synchronization patterns. Experiments on multiple datasets show strong performance in both forgery detection and generator attribution.

**Compliance With Llm Reviewing Policy:**

Affirmed.

**Final Justification:**

The rebuttal has addressed most of my concerns, and I'm not opposed to acceptance.

**Key Questions For Authors:**

I am curious about the evaluation details on LipSync-A. Since LipSync-A contains videos generated by multiple models, how does the proposed LipDA perform specifically on results generated by state-of-the-art LipSync models, such as OmniSync and InfinityTalk? Additionally, could the authors provide more failure analysis to better understand the limitations of the method?

**Strengths And Weaknesses:**

Strengths

1. The research topic is valuable, as detecting LipSync forgeries is an important aspect of AIGC safety and security.

2. The idea of leveraging AU for forgery detection is intuitive and well-motivated, and the experimental results support the effectiveness of the proposed approach.

Weaknesses

1. The applicability of the method may be limited to LipSync-style manipulations. Modern audio-driven video generation models (e.g., Wan-S2V) generate the entire video from scratch, rather than modifying only the lip region. In such cases, head pose and lip motion may remain consistent, which could potentially reduce the effectiveness of the proposed method.

2. The design of the Attribution Classifier appears somewhat questionable. It is formulated as a supervised classification task, whereas in real-world scenarios the number of possible fake sources is essentially unbounded. An unsupervised clustering formulation might therefore be more appropriate.

---

> ### Author Rebuttal · Authors · 2026-03-31
>
> Dear Reviewer #JKgT, we sincerely thank you for your precious time and valuable comments! We are encouraged by your recognition of our **research value** and **effective approach**. We sincerely hope the following clarifications and new analyses can address your concerns.
>
> ___
>
> **W1:** Applicability to Full-Video Generation Models
>
> **A1:** Thank you for this constructive suggestion. **We respectfully clarify that even for full-video generation models like Wan-S2V [1], the lip-head inconsistency persists**.
>
> - The key is not whether the generator simultaneously produces lip and head, but whether it correctly models the conditional dependence between them. As analyzed in Appendix B, the mapping {A,I}→L is near-deterministic, strictly constrained by phoneme-viseme consistency, while {A,I}→P is high-dimensional and one-to-many. Advanced LipSync generation models decompose this into two independent sub-problems to reduce optimization difficulty, losing the lip-pose conditional dependence. Specifically, for full-video models such as Wan-S2V, Sonic, and KDTalker, training objectives are dominated by lip-audio synchronization losses, with no explicit supervision enforcing lip-head coordination.
> - Table 4 demonstrates that LipDA achieves above **84%** accuracy on unseen Sonic and KDTalker videos. Following your valuable suggestion, we further evaluated 85 videos generated by the commercial Wan-S2V model and achieved **88.24%** detection accuracy. We will include additional AU analysis examples and extended results on full-video generation models in our revision.
>
> [1] Wan: Open and Advanced Large-Scale Video Generative Models. arXiv2025.
>
> ___
>
> **W2:** Supervised Attribution Design
>
> **A2:** Thanks so much for this constructive comment!
>
> - **Open-set attribution is an important long-term goal.** We agree that the number of potential forgery sources is unbounded in practice, but purely unsupervised clustering methods currently suffer from instability and pseudo-label noise accumulation, making their outputs unreliable for forensic investigators who require interpretable results.
> - The key design choice lies in how to define the label space. Following the established LipSync survey taxonomy, we categorize generators into 7 architectural families. Generators within the same family share fundamental synthesis mechanisms, which **provides reasonable forward compatibility for emerging LipSync techniques**.
> - We conducted new experiments on generators excluded from training and **found that unseen models can be correctly attributed to their corresponding families with high accuracy**: OmniSync at **90.9%** and KDTalker at **97.3%** to the Diffusion family, and EAT [2] at **99%** to the Transformer family. Supervised classification is not a perfect solution for open-set attribution, but it provides a practical and scalable base for LipSync attribution. We will include a dedicated discussion on open-set attribution as a promising future direction in our revision. Thank you again for your truly inspiring suggestions!
>
> [2] Efficient Emotional Adaptation for Audio-Driven Talking-Head Generation. CVPR2023.
>
> ___
>
> **Q1:** Evaluation Details and Failure Analysis
>
> **A3.1:** LipDA is trained on LipSync-A, comprising 1,614 real videos and 2,312 fake videos generated by 10 generators spanning diverse architectural paradigms, including Hybrid Transformations (X2Face, 89), Keypoint-Driven (TPSM/LIA/FaceVid, 1,245), Statistical Models (Dinet, 134), GAN-based (MakeItTalk/Wav2Lip, 297), VAE-based (SadTalker, 206), Transformer-based (IP-LAP, 210), and Diffusion-based (DreamTalk, 131). We evaluate unseen SOTA generators under a strict zero-shot protocol. For OmniSync, we sample 100 videos from its official dataset. For Sonic, KDTalker, and InfiniteTalk, we generate 300 videos each using official pretrained weights, with source identities from LaPa and driving audio from VCTK. Wan-S2V is evaluated using 85 commercially generated videos. Corresponding real videos are drawn from AVLips. **LipDA achieves strong zero-shot detection across all five unseen SOTA models, with OmniSync reaching 95.5%**.
>
> |      | OmniSync | Wan-S2V | Sonic | KDTalker | InfiniteTalk |
> | ---- | -------- | ------- | ----- | -------- | ------------ |
> | ACC  | 95.5     | 88.2    | 87.8  | 84.6     | 78.8         |
>
> **A3.2:** We identify three primary failure modes, which we note are shared limitations across all evaluated baselines. (1) complex camera motion with a stationary speaker, where minimal head movement produces weak lip-head inconsistency signals; (2) persistent downward gaze rendering the lip region invisible, depriving the model of effective lip input; (3) prolonged facial occlusion causing unstable landmark extraction and noisy pose features. These cases reveal limitations when lip visibility is restricted or head motion signals are inherently sparse. We will include detailed failure case visualizations in the revised manuscript.

---

> > ### Author Rebuttal · Reviewer_JKgT · 2026-04-03
> >
> > I appreciate the author's rebuttal, which addresses most of my concerns, and I'm not opposed to acceptance.
> >
> > Although the proposed method performs worse on SOTA fake videos (e.g., 88.2% of Wan-S2V and 78.8% of InfiniteTalk) than previous ones (e.g., 95.96% on average on LipSync-A), LipDA still shows substantial enhancement against related fake video detection methods (according to Table 2). That means today's video generation models are becoming more and more powerful, and thus raises the need for upgraded forgery detection methods. I think including those detailed analyses will largely enhance this paper.
> >
> > BTW, I have recently noticed a paper named AUHead[1], which also leverages the AU features for talking head generation. I am curious whether videos generated using AU features could bypass LipDA detection and thereby achieve a jailbreak.
> >
> > ---
> >
> > [1] Lyu, Jiayi, et al. AUHead: Realistic Emotional Talking Head Generation via Action Units Control."

---

> > > ### Author Response · Authors · 2026-04-03
> > >
> > > Thank you once again for your time, attention, and valuable feedback to improving our work. We are encouraged that our rebuttal has addressed most concerns.
> > >
> > > LipDA achieves strong zero-shot detection on unseen SOTA generators, demonstrating good generalizability. We agree with your insightful framing that as generative models grow increasingly powerful, detection becomes more challenging, resembling an arms race. We will add detailed analyses of per-model performance trends and their implications to further strengthen the paper.
> > >
> > > AUHead's AU conditioning is designed solely to control emotional expressiveness, with its training objective focused on visual quality and audio-lip synchronization. In AUHead, head motion and AU-driven lip motion are generated independently, with no mechanism to enforce their coordination. LipDA detects the broken temporal coupling between lip motion and head pose, a property inherent to real speech. Therefore, LipDA generalizes naturally to AUHead-generated videos. More tellingly, AUHead's own quantitative results confirm that AU conditioning does not enhance lip-head coordination, as SyncNet scores decrease from base models to AUHead variants in AUHead's Table 3.
> > >
> > > Unfortunately, AUHead's official repository currently contains no released code or pretrained weights, making direct experimental validation infeasible at this stage. We will include a dedicated discussion of AUHead as an emerging threat model in the revision, and will conduct more evaluation to empirically verify LipDA's robustness against AU-conditioned generation.

---

### Decision · Program_Chairs · 2026-04-30

**Decision:**

Accept (regular)

**Comment:**

The paper received positive scores of 4/4/4/4.

After the rebuttal, all concerns were addressed, and no further issues were raised. All reviewers consistently recommended Weak Accept.

After carefully reading the submission, the reviews, and the discussion, the reconmendation is: Accept.